# Learning Case Study of a Shallow-Water Model to Assess an Early-Warning System for Fast Alpine Muddy-Debris-Flow

**Antonio Pasculli** [1,2,*], **Jacopo Cinosi** [1], **Laura Turconi** [3] **and Nicola Sciarra** [1]

[1] Department of Engineering and Geology, University of "G. D'Annunzio", Chieti-Pescara, 66013 Chieti, Italy; jacopocinosi@yahoo.it (J.C.); nicola.sciarra@unich.it (N.S.)

[2] INDAM Research Group GNCS, National Institute of Advanced Mathematics, National Group of Scientific Computing, University of "G. D'Annunzio", Chieti-Pescara, 66013 Chieti, Italy

[3] National Research Council (CNR), Research Institute for the Hydrogeological Protection (IRPI), 10135 Torino, Italy; laura.turconi@irpi.cnr.it

[*] Correspondence: a.pasculli@unich.it; Tel.: +39-0871-355-6159

**Abstract:** The current climate change could lead to an intensification of extreme weather events, such as sudden floods and fast flowing debris flows. Accordingly, the availability of an early-warning device system, based on hydrological data and on both accurate and very fast running mathematical-numerical models, would be not only desirable, but also necessary in areas of particular hazard. To this purpose, the 2D Riemann–Godunov shallow-water approach, solved in parallel on a Graphical-Processing-Unit (GPU) (able to drastically reduce calculation time) and implemented with the RiverFlow2D code (version 2017), was selected as a possible tool to be applied within the Alpine contexts. Moreover, it was also necessary to identify a prototype of an actual rainfall monitoring network and an actual debris-flow event, beside the acquisition of an accurate numerical description of the topography. The Marderello's basin (Alps, Turin, Italy), described by a 5 × 5 m Digital Terrain Model (DTM), equipped with five rain-gauges and one hydrometer and the muddy debris flow event that was monitored on 22 July 2016, were identified as a typical test case, well representative of mountain contexts and the phenomena under study. Several parametric analyses, also including selected infiltration modelling, were carried out in order to individuate the best numerical values fitting the measured data. Different rheological options, such as Coulomb-Turbulent-Yield and others, were tested. Moreover, some useful general suggestions, regarding the improvement of the adopted mathematical modelling, were acquired. The rapidity of the computational time due to the application of the GPU and the comparison between experimental data and numerical results, regarding both the arrival time and the height of the debris wave, clearly show that the selected approaches and methodology can be considered suitable and accurate tools to be included in an early-warning system, based at least on simple acoustic and/or light alarms that can allow rapid evacuation, for fast flowing debris flows.

**Keywords:** numerical modelling; muddy flow; shallow water; parametric sensitivity analyses; rheological laws; GPU approach

## 1. Introduction

Mud-flows are natural phenomena, appearing as waves with a steep front and a shape resembling that of debris-flows, but consisting of a muddy slurry that contains much less boulders and much less granular material. Nevertheless, mud-flows can cause severe damage to human settlements and infrastructures and may produce many casualties, as much as debris-flows do. Many articles are present in bibliography on the subject. In the following, just a few examples concerning the physical based approach selected for our paper, are given [1–8]. Within the early warning system, other classical types of methodology have been proposed, in particular on rainfall thresholds, discriminated in deterministic and probabilistic approaches [9]. On the other hand, our research aims

to be a further in-depth contribution, also regarding our previous paper [8], based in particular on experimental measurements and geo-informatic approach coming from different and multiple sources, also through a detailed parametric sensitivity analyses, varying some leading parameters, difficult to be directly measured. The related assumed values, although reasonably justifiable from the physical point of view, were essentially used to calibrate the simulations. Moreover, they were used to verify the feasibility of a robust advanced numerical and experimental approach for the development of a simple "early debris warning system", mainly consisting of acoustic and/or intermittent light signals. In the paper, a learning-test case was examined for which reliable experimental measurements were available. The calibration was, therefore, carried out on the basis of the data of a single event. Obviously, to calibrate even a simple early warning specific to the site under study, an analysis of several cases would be necessary in order to obtain a representative statistics of the specific geo-mechanical parameters of the territory under study. However, in this step of the research activity, the applicability of the method in question is being explored, in particular if the computational time, possibly further reduced by more powerful hardware, can allow a prompt alert of the occurrence of dangerous debris flow phenomena. From this point of view, the study can also be applied to other similar cases as a feasible methodology. Accordingly, a mathematical-numerical modelling based on real-time rainfall measurements, almost simultaneous with the possible damaging event, should strictly meet two requirements: to be as accurate as possible and, at the same time, have a low computational cost. Moreover, an adequate description of rainfall distribution within the territory should be mandatory. The best way to acquire this information would be the use of radar data, continuously distributed within both over time and space. On the other hand, a network of punctual rain gauges is much more commonly available, often distributed according to different criteria of opportunity. The calibration of the pluviometric network and the consequent transmission of the data to a central hardware-software, used for their processing, are also necessary topics to be considered. In order to study this kind of phenomena, the use of simple models (for example, 1D approach), usually, can solve only a few aspects of the problem. On the other hand, the category known as "complex conceptual models" assumed that most of their parameters could be defined from the physiographic characteristics of the basins. In this field, advanced utilized approaches are based on the balance equations of physics, pursued by the *Computational Fluid Dynamics* (CFD). The CFD approach relies on solving Navier–Stokes flow equations [10,11], supplemented by experimental laws that define the parameters related to the mass transport of material, the rate of erosion (for instance: [12]), etc. Besides the most common "grid-based methods", another category of approaches was proposed: the mesh-less solution of the differential equation. Among others, *Smoothed Particle Hydrodynamics* (SPH) appears to be quite well established ([6,13,14] among many others). The *Reduced Complexity Models* (RCM), to which, for example, *Cellular Automata* (CA) approach belongs [15–17], represents an important alternative to the CFD, in particular in order to predict morphological changes, within large area and over relevant time scale (climate evolution as well), both at reach and at catchment scale. Therefore, for the study of the phenomena considered in this paper, involving short time scale, a suitable compromise could be found in approaches capable to simplify CFD technique, like the *Shallow Water* approach, widely used in a simplified but effective way (among many others: [18]) or in more sophisticated way as it was implemented into the RiverFlow2D code [19] that was selected to perform the simulations discussed in the following sections. The mathematical–numerical approach of RiverFlow2D is based on the *Shallow Water* models called *classic*, aimed at simulating flows over small bottom slopes. On the other hand, Van Emelen et al. [20] compared the *classical* approach to a modified *Shallow Water* model accounting for the incidence on the flow of the steep slopes, such as those that characterize the topography of basins similar to the Marderello's site. For rapid transient events, similar to the debris flows we have simulated, it was shown that nearly no difference appears between classical and modified models. For steady uniform flows, however, differences are visible for bed angles higher than 10°, indicating that

classical shallow flow equations are not suitable for uniform flows on steep slopes. On the other hand, the main purpose of our work was to verify the suitability of this type of modeling precisely to predict the arrival time and the maximum peak value of fast debris flows, in general, not uniform. At this stage of the research, the erosion of both the river bed and banks, with the consequent production and transport of sediments, was not explored. Accordingly, in this step of the research, no deep discussion about grain distribution and topics of the related measurements, are given. Actually, the parameters characterizing the selected rheological laws are expected to include also the effects of the presence of sediments. Another important feature, included in RiverFlow2D as well, is the possibility to include frequent wetting and drying phenomena, induced also by rainfall variability, which may lower soil mechanical strength (for example, for pyroclastic soil: [21,22]), triggering possible landslides occurrences and, accordingly, morphological variation of the territory. On the other hand, important phenomena like turbulence [23,24], are missed or not accurately included into the present version of the code. The simulation of debris flow phenomena was performed by a 2D Finite Volume Method, based on the Godunov–Riemann Shallow-water method. After the selection of a possible available mathematical–numerical approach, the choice of a suitable test case was consequential. To this purpose, Italian territories, similar to many areas around the world affected by this particular type of phenomena, due to both natural and anthropogenic causes, were considered. In particular, in the Alps, debris flows occur with a frequency high enough to create serious hazards to human settlements. In 1994, a small creek located in the North-West Italian Alps that presented a very high debris flows occurrence was thus selected by the National Research Council (CNR), Research Institute for the Hydrogeological Protection (IRPI) CNR-IRPI of Turin for the installation of a debris flow monitoring system consisting of a hydro-meteorological network, based on seven rain gauges (only five of which were considered in this paper because the only ones embedded inside the area of influence of the phenomenology under study, identified through the Voronoi's polygonization) in different portions of the Marderello's basin [25]. The monitoring equipment was recently extended along an alluvial fan with one ultrasonic water level sensor, two video-cameras, and four vertical geophones [26]. Accordingly, the Marderello's basin and the muddy debris flow event that occurred and monitored on 22 July 2016, were identified as a typical test case, well representative of the mountain context and the phenomena under study. First of all, a morphological digital model of the basin under study, based on a $5 \times 5$ m DTM, was created, implemented and checked as inputs. The inflow boundary conditions, deriving from the hydrogram consisting of values measured at five monitoring gauges, were considered. Several parametric analyses were performed in order to individuate the best values fitting the measured flow-height of the muddy debris over time. To this purpose, different available rheological options such as clear water, *Coulomb-Turbulent-Yield*, *Turbulent*, and *Full Bingham* were selected and tested. It is worth noticing that the *Coulomb-Turbulent-Yield* model includes also the bed slope angle (Equation (6)).

Accordingly, the main purpose of our paper was a further contribution to the discussion of the feasibility of the selected numerical approach to simulate real events well representative of debris-like phenomena, occurring in monitored situ. The contemporary fulfilment of two apparently conflicting requirements, but both necessary for an early-warning system, such as the accuracy of the selected mathematical model and the low computational time required to perform numerical simulations, was discussed and explored, as well. Mathematical models based on the classical balance principles of the physical quantities of interest, such as the shallow water approach, selected for this work, certainly show greater flexibility and generalization of their use than the possible use of less complex empirical models which are still valid [9,27,28], but whose applicability is limited to the typologies of phenomena for which they have been proposed and, accordingly, cannot have a general validity. On the other hand, models based on CFD require a great amount of computing time. However, this work attempts to demonstrate that with the use of the GPU tool, advanced mathematical models could be considered for developing early warning systems as well.

Accordingly, another important contribute was a deep investigation, through a sensitivity analyses, aimed at acquiring further suggestions about the numerical values of some important characterizing parameters and about the necessary improvement of existing Shallow-Water approaches.

## 2. Materials and Methods

### 2.1. Geographical and Geological Setting of the Site Selected as Representative Test-Case (Marderello's Basin Turin, Italy)

The Marderello is a left tributary of the Cenischia valley (near Novalesa, Piemonte region, NW Italian Alps). This small basin (6.61 km$^2$) has elevations ranges between 3538 m (Mt. Rocciamelone peak) and 900 m above sea level (a.s.l.) (fan apex), with a total drop of about 2638 m in 4 km (slope 65%). A North-South oriented fault system resulted in a complex network of rock joints and cracks [26]. The bedrock of the Cenischia valley belongs to the Tectono-metamorphic units of Rocciamelone and Puys–Venaus (Deep Oceanic Units; Servizio Geologico d'Italia, 1999). The Rocciamelone unit caps the highest area above 2600 m; it is made of a succession of calcschist and silicate marble. Ophiolites are tectonically interposed in the basal part of silicate marble, which is found near Cà d'Asti Refuge (2854 m). Clayey-arenitic schists are widely overlain by deep-seated slope collapse deposits and colluvium. The bulk of degradation processes develops between 2400 and 1300 m a.s.l., primarily induced by high relief energy. The climate features reflect a transitional situation, in which geographical and orographic factors play a key-role in local weather conditions. The average annual precipitation, including both rainfall and snowfall amounts to 820 mm, the majority of which (62%) occurs in autumn and spring [26]. A thick snow-mantle (1–3 m) normally caps the slopes above 2500 m a.s.l. from the middle of October to the end of June. During this period and above this elevation, soil and superficial detrital-covers frequently are in wet conditions and this exposes the slopes to erosion processes, landslides and debris-flows initiation, chiefly induced by summer rainstorm. In the Marderello catchment, the debris-flow source areas are quite difficult to ascertain. In most cases the whole catchment provides the solid material necessary to form the mixture, but the debris cover in the channel and along the banks constitutes the main detrital sources. Therefore, the sediment delivery is characterized by different and not homogeneous processes of erosion and deposition; indeed, the source areas location and their distance from the fan can change the debris-flow initiation mechanism and runout distance. Furthermore, from the start and throughout its evolution, the phenomenon undergoes noteworthy modifications according to the different variables in play each time and to the geomorphologic conditions. In particular, the topographical steps due to the morphological and structural glacial history of the Marderello Torrent determine changes in the composition of the solid–liquid mixture. In the torrent intermediate section, the succession of canyons and waterfalls operates as a "filter", significantly reducing the grain size and the solid volume percentage in the mixture. This effect imposes significant variation to the sediment transport typology: from a stony debris-flow initiating in the upper basin the phenomenon evolves in a muddy debris-flow or a mud-flow on the alluvial fan.

### 2.2. In Situ Measurement Devices

Figure 1 shows the localization of 5 selected pluviometric stations (colored circles) and the ultrasonic transducer gauge (blue star) utilized to measure, respectively, the rainfall intensities and the flow depth related to the event under study.

The resolution of the installed tipping bucket rain gauge is about 0.2 mm for each commutation of the bucket when the rainfall intensity reaches the value of 100 mm/h. The hydrometer was collocated under a small bridge. The instrument's measuring capability of the distance from the water surface ranges from a minimum of 0.8 m up to 16 m, fully compatible with the distance of about three meters of the hydrometer from the bottom of the creek, with a sensibility of 0.5 cm and an accuracy of $\pm1$ cm. The instrument takes also into account the variation of the sound velocity due to the air temperature variation.

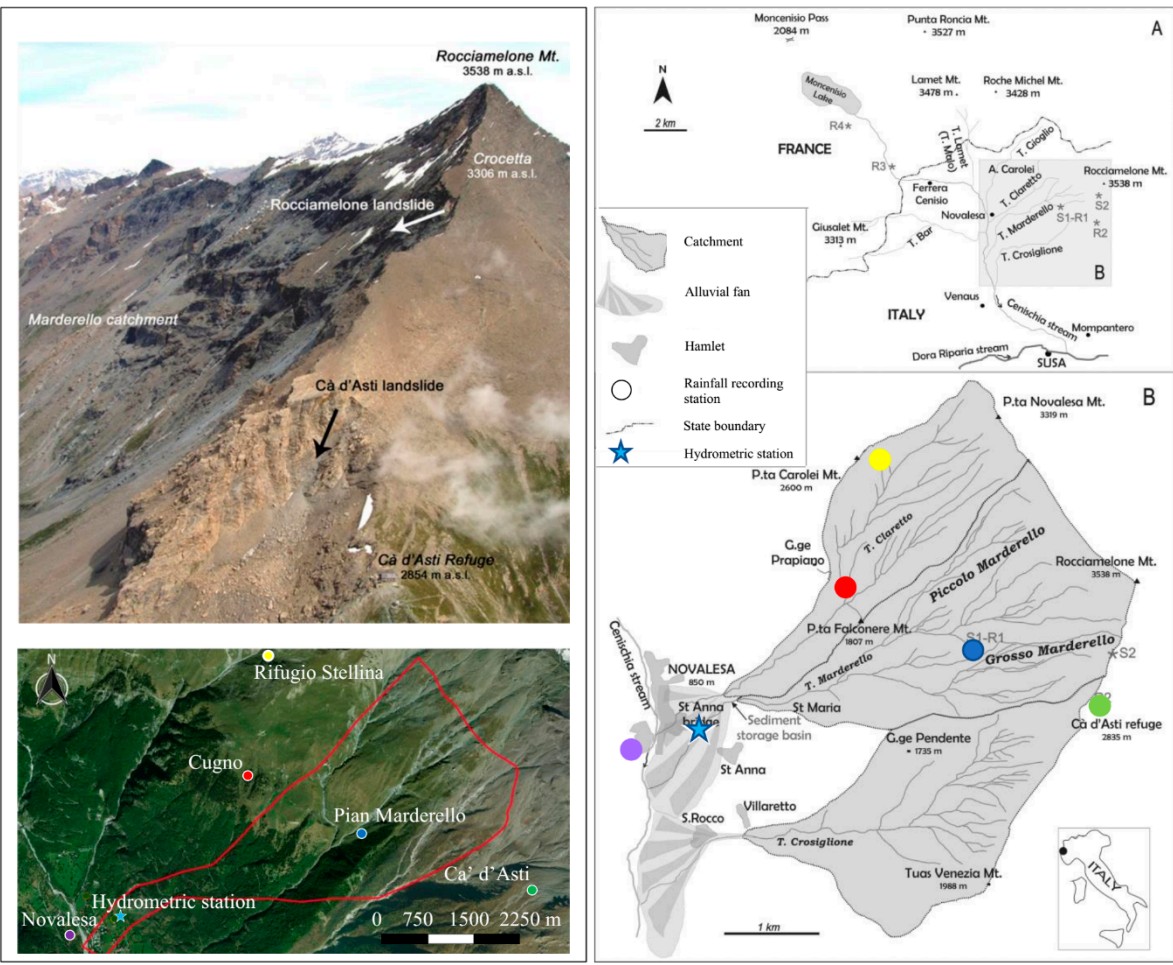

**Figure 1.** Map of the Marderello site; Pluviometric and Hydrometric Stations location.

### 2.3. Measured Data

The available experimental data covered the rainfall and the hydrometry of the Marderello torrent from 21 July 2016 (19:00) to 22 July 2016 (8:50) with a cadence of one minute (Figure 2). However, from the inspection of the plots the rainfall event at 19:30–19:50, occurred on 21 July 2016, characterized by a maximum rainfall intensity spike of about 30–35 mm/h(Ca' d'Asti and Marderello) with a total of 6 mm rainfall cumulate, did not induce any relevant variation of the hydrometry. By consequence, the starting point for the numerical simulation was selected, essentially, at 5:40 of 22 July 2016. From the inspection of the plots (Figure 2), it appears that, for the morphology of this basin, a rainfall intensity threshold greater than about 30–40 mm/h may generate a wet debris flow at the selected hydrometric station site. The flow depth measured at the hydrometric gauge observed a constant value around 0.38 m and, after the mean spikes, the measured flow depth assumes values ranging from 0.28 to 0.35 m. Some frames (Appendix A Figure A1) extracted from movies of the 22 July 2016 event, obtained by video cameras shooting the upstream channel and located upstream the hydrometer, show that the mud debris flow was anticipated and followed by less intense flow of water. The video camera was placed on the bridge (Via Roma, Novalesa, Turin, Figure 3c) in the middle of which the hydrometer was positioned. Reasonably, the measured flow depth was also affected by brushwood deposit. Reasonably, these kinds of material were also partially carried away by the flow of water, as it may be inferred observing that after the peaks of the mud flow, the flow depth decreased slightly and recovered the level just later (Figure 2). These details are not essential for the purposes of this paper, at regarding early-warning aspects for which just an estimate

of the time arrival, possibly slightly in advance, and of the height of the mud debris front, desirably slightly higher than the real one, would be required from a safety point of view.

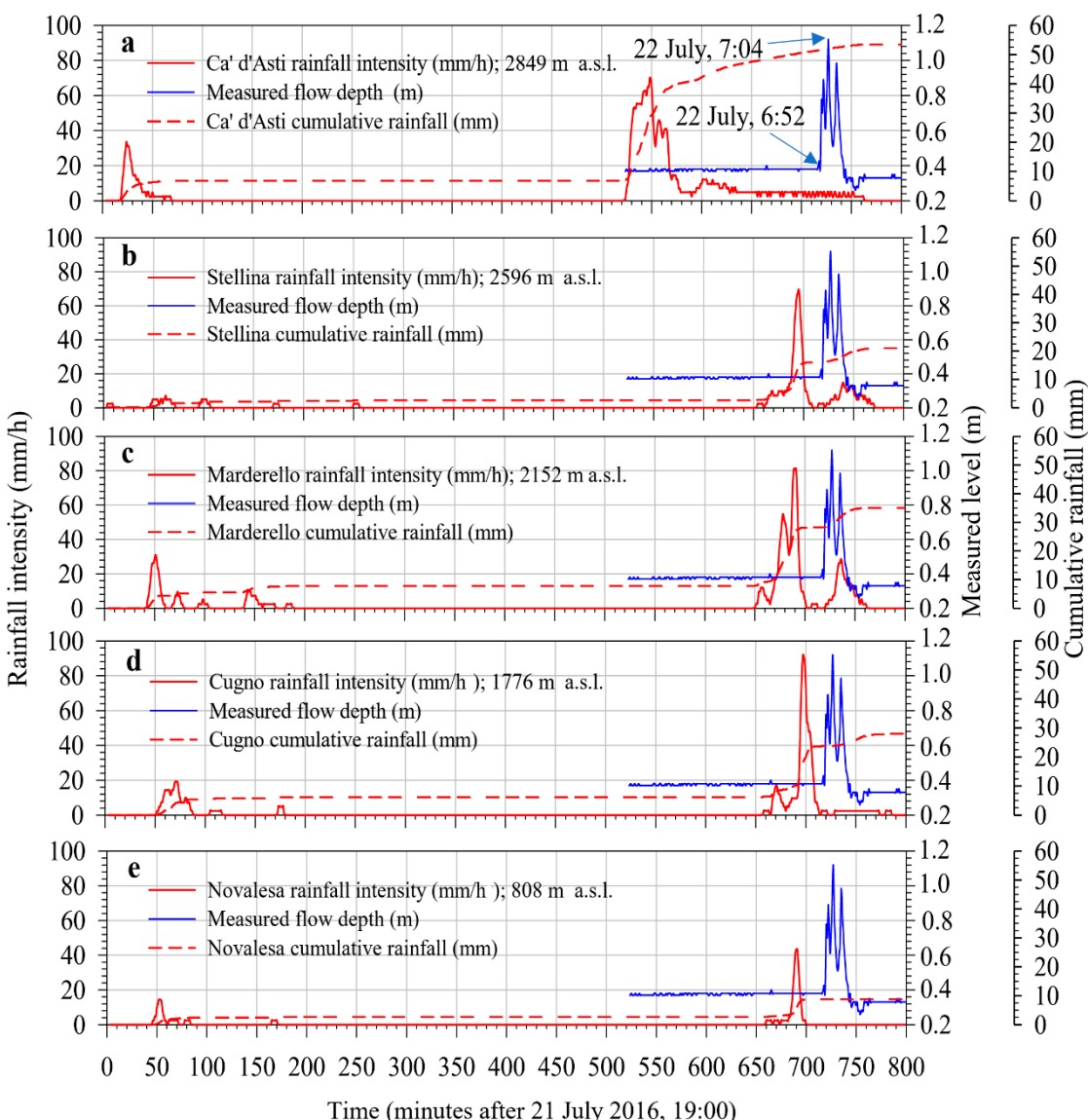

**Figure 2.** Rainfall time histories, measured flow depth and cumulative rainfall of the available five pluviometers, from 21 July 2016, 19:00 to 22 July, 8:50; (**a**): Ca' d'Asti 2849 m a.s.l.; (**b**) Rifugio Stellina 2596 m a.s.l.; (**c**) Pian Marderello 2152 m a.s.l.; (**d**) Cugno 1776 m a.s.l.; (**e**) Novalesa 808 m a.s.l.

However, some numerical simulations predicted a residual flow before the mean mud flow peaks arrival and, after the peaks, excessive flows compared to measured flow level were obtained in all simulations as will be discussed in the following sections.

The choice of rheological models essentially depends on the characteristics of the debris flow mixture. However, the testing of the models implemented in the selected code and how effective they are in predicting experimental results are among the aims of the paper. The different available options do not require the granulometric structure of the muddy debris flow under study, nevertheless, just for completeness reasons, some data, estimated by CNR of Turin, are provided. Excluding blocks and ultra-metric boulders, the values of D15, D50, and D70 (the intercepts for 15%, 50% and 70% of the cumulative mass) range, respectively, between 0.003 and 0.08, 0.04 and 17, and 0.065 and 35 mm, while the density ranges from 1992 to 2047 kg/m$^3$.

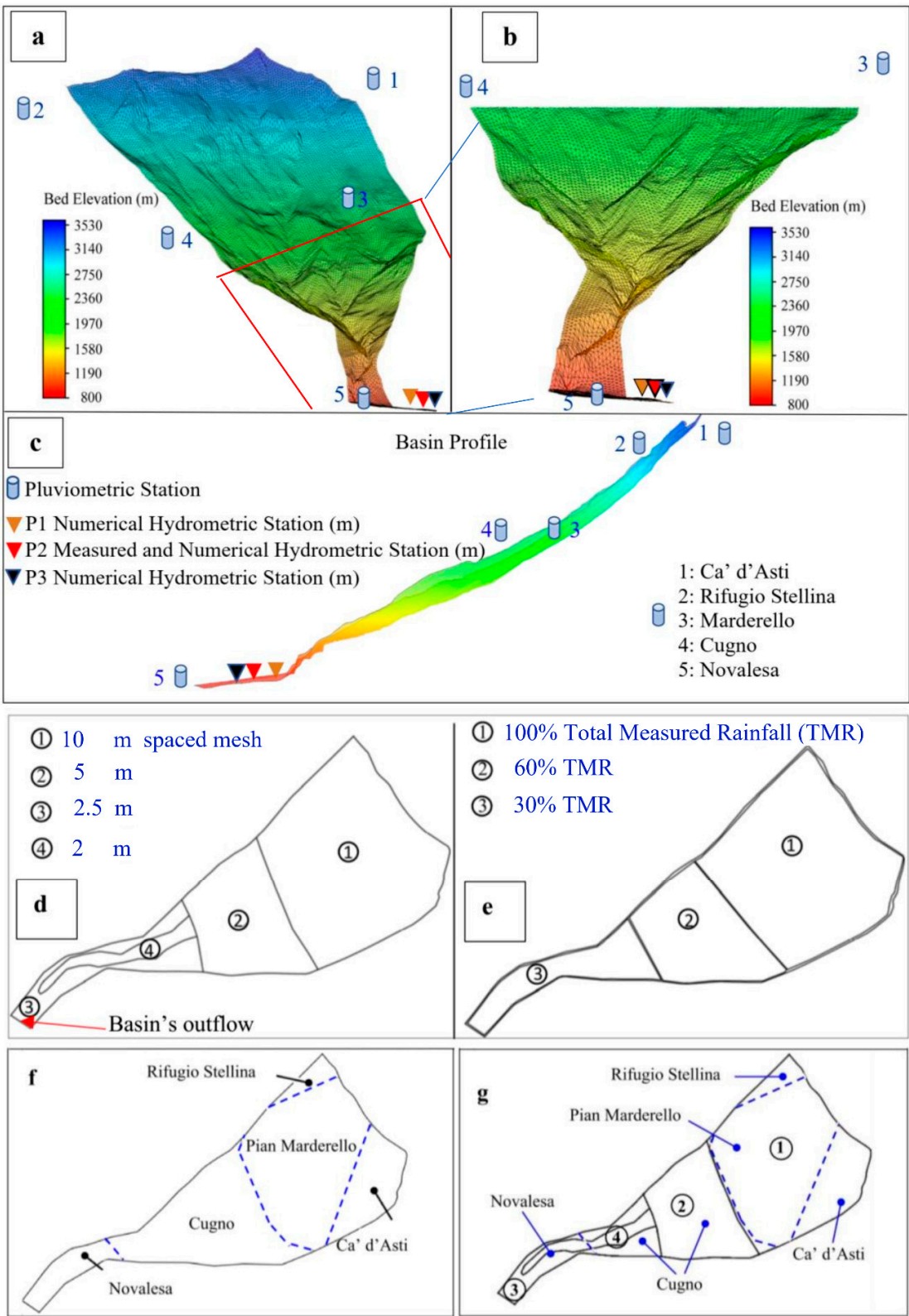

**Figure 3.** Numerical meshing of the selected spatial domain of the Marderello basin; (**a**–**c**) sketch of the collocation of the of the pluviometric and hydrometer gauges proxies and the position of the video camera; pluviometer gauges altitudes (m a.s.l.): (1) Ca' d'Asti 2849; (2) Rifugio Stellina 2596; (3) Pian Marderello 2152; (4) Cugno 1776; (5) Novalesa 808; Hydrometer 839; (**d**) basin area subdivisions based on variable average mesh width; (**e**) basin area subdivisions based on different rainfall intensity areas; (**f**) an adaptation of the Thiessen's polygonization to the morphology of the Marderello's basin; (**g**) superposition of the resulting polygonization of rainfall distribution and spatial domain discretization.

### 2.4. Selected Mathematical–Numerical Approach

Among many available computer codes, RiverFlow2D, 2017 year version, based on the Finite Volume Method (FVM, [19]) as numerical solver for free surface flow models and on the SMS tool, a general purpose Graphical User Interface (GUI), supplied with the code, as pre and post processor, were selected. The Depth-Averaged 2D shallow-water approach [29–32], adopted by the selected software, is based on the assumption that the vertical depth of the flow is negligible compared to the other flow's dimensions. This is an acceptable compromise between the need to adopt an approach as realistic as possible and the need to avoid long computation times such as those required by 3D approaches. In order to also consider bed level jumps (considered as source terms in the related differential equations), due to the geomorphology of real complex landform, the code includes the *Augmented approximate Riemann solvers* approach [31]. In addition, the selected code allows numerical parallel simulations based on the Graphic Processing Unit (GPU) card, which is able to run many times faster than in single processor computers. To perform the content of the paper, among the other available software's options, the Hydrodynamic and the Mud-Debris flow models were selected. In the version of the code selected to develop this work, important parameters such as the Manning coefficient, the debris flow density, its internal viscosity, and other important physical-geo-mechanical parameters are considered constant throughout the transient. Of course, this approach constitutes a weakness of the mathematical–numerical tool. Nevertheless, the new 2020 version of the code, www.hydronia.com (accessed on 15 February 2021), not used yet for the research activity discussed in this paper, attempts to solve some drawbacks of the previous version. In the following, a summarized description of the utilized mathematical and numerical approaches, freely adapted from [19,30–32], in which many more details are reported, is briefly discussed.

2.4.1. Hydrodynamic Unsteady Flow Models

The following is the resulting system of the coupled partial differential equations, based on the water volume conservation, water momentum conservation and shallow water assumption [30–32]:

$$\frac{\partial \boldsymbol{U}}{\partial t} + \frac{\partial \boldsymbol{F}(\boldsymbol{U})}{\partial x} + \frac{\partial \boldsymbol{G}(\boldsymbol{U})}{\partial y} = \boldsymbol{S}(\boldsymbol{U}, x, y) \tag{1}$$

$$\boldsymbol{U} = \left(h,\ q_x,\ q_y\right)^T;\ \boldsymbol{F} = \left(q_x,\ \frac{q_x^2}{h} + \frac{1}{2}gh^2,\ \frac{q_x q_y}{h}\right)^T \quad \boldsymbol{G} = \left(q_y,\ \frac{q_x q_y}{h},\ \frac{q_y^2}{h} + \frac{1}{2}gh^2\right)^T \tag{2}$$

$$\boldsymbol{S} = \left(0,\ \frac{p_{bx}}{\rho} - \frac{\tau_{bx}}{\rho},\ \frac{p_{by}}{\rho} - \frac{\tau_{by}}{\rho}\right)^T \tag{3}$$

where: $h$ is the water depth; $q_x = h\,u$ and $q_y = h\,v$ the unit discharges, resulting from the *shallow water* approach; $(u, v)$ the depth averaged components of the velocity vector $u$ along, respectively, $x$ and $y$ coordinates; $g$ is the gravity acceleration; $gh^2/2$ the flux obtained after assuming a hydrostatic pressure distribution in every water column, as common practice in *shallow water* models. The source term $\boldsymbol{S} = (\boldsymbol{U}, x, y)$ incorporates the effect of pressure forces $p_{bx}$ and $p_{by}$ over the bed and the tangential forces $\tau_{bx}$ and $\tau_{by}$. Moreover, $p_{bx}/\rho = ghS_{0x}$ and $p_{by}/\rho = ghS_{0y}$, while $S_{0x} = -\partial z/\partial x$ and $S_{0y} = -\partial z/\partial y$ are the bed slopes of the bottom level $z$. The tangential forces are characterized by selected rheological laws discussed briefly in the following sub-sections.

2.4.2. Mud and Debris Flow Model (RiverFlow2D MD Module)

The adopted mud-debris fluids models (based on [33]), typically non-Newtonian, are considered as hyper-concentrated water and sediment mixture affected by stop and go mechanisms and regard bed and internal friction for free-surface flows, ranging from clear water to hyper-concentrated mixtures of sediments. In addition, simulation of dry–wet

and wet–dry transitions is allowed. The model involves the following assumption: (a) the flow is confined to a layer that is thin compared to the horizontal scale of interest; (b) the water and sediments mixture is assumed to be an homogeneous single phase with constant properties, e.g., density, yield stress etc. and is described by continuum approach, governed by Equations (1)–(3); the liquid and the solid phase velocities are assumed to be the same; (c) the river bed does not erode. An example of a different approach is discussed in [34]. The single-phase MD rheological formulation in RiverFlow2D MD accounts for different tangential forces friction terms, included in a same mathematical expression, that represent a variety of hyper-concentrated non-Newtonian fluids. For the simulations performed and discussed in this paper, some available options were selected [35], each differing from the others through the mathematical expressions assumed by the tangential forces $\tau_{bx}$ and $\tau_{by}$. In order to explore a simplified approach, the *clear-water* option was exploited as the first step of the study, based on the Manning friction law:

$$\tau_{bx} = S_{fx} \equiv \frac{n^2 u \sqrt{u^2+v^2}}{h^{4/3}}; \qquad \tau_{by} = S_{fy} \equiv \frac{n^2 v \sqrt{u^2+v^2}}{h^{4/3}} \qquad (4)$$

where $n$ is the Manning coefficient related, in some way, to the roughness coefficient.

Then, other rheological models were selected aimed at describing in more realistic way the muddy-flow behaviour, summarized in the following.

- Full-Bingham (FB): for this option $\tau_{b\ x} = \tau_{FB\ x}$; $\tau_{b\ y} = \tau_{FB\ y}$ are the solutions of the following two equations:

$$\begin{cases} 2\tau_{FB\ x}^3 - 3(\tau_{yield} + 2\tau_{\mu\ x}) \cdot \tau_{FB\ x}^2 + \tau_{yield}^3 = 0 \\ 2\tau_{FB\ y}^3 - 3(\tau_{yield} + 2\tau_{\mu\ y}) \cdot \tau_{FB\ y}^2 + \tau_{yield}^3 = 0 \end{cases} \qquad (5)$$

where $\tau_{\mu x} = \mu_B \frac{u}{h} \equiv \mu_B \frac{q_x}{h^2}$ and $\tau_{\mu y} = \mu_B \frac{v}{h} \equiv \mu_B \frac{q_y}{h^2}$, while $\tau_{yield}$ is the threshold value of the yield stress beyond which the material starts flowing. Once flowing, the movement is characterized by the Bingham viscosity $\mu_B$ of the mixture.

- Turbulent (Turb): in this case: $\tau_b = \rho c_f(u^2 + v^2)$ where $c_f$ is a friction coefficient [33–35].
- Coulomb-Turbulent-Yield (CTY): this was the most applied rheological law for this paper in which

$$\tau_b = \rho c_f(u^2 + v^2) + \min(\tau_{yield},\ g\rho h \cdot \cos\theta \cdot \tan\theta_b) \qquad (6)$$

where $\theta$ is the bed slope angle and $\theta_b$ is the friction angle of the solid material. An important parameter is the Manning's coefficient $n$, whose value should be selected by the user, that usually accounts for the effects of bed roughness, internal friction and variations in shape and size of the channel cross section, obstructions, river meandering.

### 2.4.3. Manning's Coefficient

The Manning's "$n$" coefficient usually accounts for the effects of bed roughness, internal friction and variations in shape and size of the channel cross section, obstructions, river meandering. However, the selected 2D numerical approach does not account for lateral friction. For these reasons, the "$n$" coefficient was partially used as a tune parameter in the calibration process in order to adjust the numerical results to measured data. The initial guessed values, employed in this paper, were based on the tables reported in [36]. In particular, most of the simulations were performed considering values ranging from 0.04 to 0.055, compatible to the characteristics of the Marderello's basin, with a few vegetation in channels and incisions, steep banks, a few trees and brush. Final simulations were carried out considering a value of 0.15, compatible with the presence, in summer, of medium to dense brush. This last value is within the recommended range for open grounds with debris [34].

### 2.4.4. Infiltration

In general, the hydrological budget resulting from the rainfall is divided into several components, among which the runoff, that causes the formation of debris flow, infiltration and evaporation. From in field inspections, it was found that there were no snow deposits that could contribute to the water balance. The evaporation phenomena, given the altitude, the temperature and the short period of the transient, can be considered negligible. Therefore, the hydrological balance of the streamed material was based only on the rainfall inventory and on the infiltration processes. Among the available options in the selected software [19], the Horton model was selected. This commonly used type of model is particularly suitable for the parametric analyses performed in this work. For this kind of infiltration model [37], the soil infiltration capacity $f_p$ (m/s) is based on the following exponential law:

$$f_p = f_c + (f_0 - f_c) \cdot e^{-kt} \tag{7}$$

where $f_0$ and $f_c$ are, respectively, the initial and final infiltration capacities, both measured in m/s and $k(\text{s}^{-1})$ is the rate of decrease in the capacity. Actually, the development of the Formula (7) is based on an experimental approach, even if a reasonable justification can be acquired by considering the temporal saturation rate of the soil compared to the intensity of the rainfall. Accordingly, the resulting runoff inventory would be determined by the balance between rainfall and infiltrated water. At the same given rainfall intensity, the minimum value of the water that could runoff is obtained when the initial conditions of the soil are dry, while the maximum value is reached when the soil is in saturation conditions. However, the related coefficients must be determined from experimental data that could be found in [38,39]. On the basis of the specific characteristics of the Marderello basin's soil, the following ranges of numerical variability of the parameters were explored: 0.05 $(\text{s}^{-1})$ $\leq k \leq 0.5$ $(\text{s}^{-1})$; $1.58 \times 10^{-6}$ (m/s) $\leq f_c \leq 5.00 \times 10^{-6}$ (m/s) silt loam; $7.06 \times 10^{-6}$ (m/s) $\leq f_c \leq 13.38 \times 10^{-6}$ (m/s), dry clay or moist loam soils with a few to no vegetation at all (minimum values) or dense vegetation (maximum values). For the simulations discussed in this paper, the same values of the infiltration parameters were selected for the whole area of the basin.

### 2.4.5. Numerical Solver

The phenomena studied in this work require the simulation of flooding of debris flows that can undergo sudden variations of the material content, transported through a territory affected by eventually discontinuous variable morphology. The numerical solvers of the "conservation partial differential equations (PDE)", resulting from the application of shallow water models, can be identified in the context of "initial and boundary value" problems, with "source terms" to which the abrupt morphological variability of the paths of the debris flows can be assimilated by a numerical point of view. However, the search for solutions must face the challenge posed by the presence and/or the occurrence of discontinuities, due to both the wet–dry state transitions of the control volume and the discontinuities of the topography. Therefore, the numerical context is that of the "Riemann Problems (RP)", which naturally appears in Finite Volume Methods, Godunov (FVM) type approaches [31]. For its solution, the "Riemann solver" [31] method was implemented into RiverFlow2D. Within this framework, quantities such as rarefaction and shock waves, generated on discontinuities, appear as "characteristics", identified by the "eigenvalues"approach, along which the PDEs are transformed into ordinary differential equations (ODE), easier to be solved than PDE. Roe's approach is also part of the RP solution [31], whereby the integral of the approximate solution of the linearized RP, on a suitable control volume, is equal to the integral of the exact solution on the same control volume. To pursue this requirement in the Roe's method, the approximate solution can be defined using an approximate Jacobian matrix of the non-linear normal flow and two approximate matrices, constructed using the Jacobian eigenvectors. In order to obtain a completely conservative method, RiverFlow2D considers the complete system, including the hydrodynamic and transport equations.

Mathematically, the complete system retains the property of hyperbolicity, implying the existence of a $4 \times 4$ Jacobian matrix for the 2D model.

Accordingly, the general underlaying numerical approach is based on the Finite-volume scheme, by the integration in a volume or grid-cell $\Omega$ using Gauss theorem:

$$\frac{\partial}{\partial t} \int_{\Omega} \boldsymbol{U} d\Omega + \oint_{\partial \Omega} \boldsymbol{E}(\boldsymbol{F}, \boldsymbol{G}) \cdot \boldsymbol{n} ds = \int_{\Omega} \boldsymbol{S} d\Omega \qquad (8)$$

where $\boldsymbol{U}$, $\boldsymbol{F}$, $\boldsymbol{G}$ and $\boldsymbol{S}$ are the vectors already described in (2), $\partial \Omega$ is the volume domain boundary, $\boldsymbol{n}(n_x, n_y)$ is the outward unit normal vector to the volume $\Omega$. Then, a piecewise representation of the conserved variables and an upwind and unified formulation of fluxes and source terms is applied [30]:

$$\frac{\partial}{\partial t} \int_{\Omega_i} \boldsymbol{U} d\Omega + \sum_{k=1}^{\text{(Number of Edges)}_i} (\boldsymbol{E} \cdot \boldsymbol{n} - \overline{\boldsymbol{S}})_k A_k = 0 \qquad (9)$$

where $\Omega_i$ is the volume of the *i-th* computational cell, whose area of the *k-th* edge face is $A_k$. Then, the approximate solution is always constructed as a sum of jumps or shocks, also involving rarefactions [31].

Optimal Time-Step Computation

The problem of identifying the optimal time-step is part of the search for the requirements to be met in order to obtaining the stability of the selected algorithms. RiverFlow2D uses an automatic implementing procedure, based on the solution of the RP [30]. In 1D simulations the time step is taken small enough in order to avoid interaction of "numerical waves" emerging from the application of this particular type of approach [31]; then the "equivalent distance" $\Delta x/2$ between neighboring meshes is introduced. In the 2D framework, considering unstructured meshes, the equivalent distance, that will be referred to as $L_i$ in each *i-th* cell, must consider the "volume" of the cell and the length of the shared edges:

$$L_i = \frac{A_i}{\max\limits_{k=1,\,NE} l_k} \qquad (10)$$

where $A_i$ is the "volume" (area in 2D) of the *i-th* cell, $l_k$ is the length of the *k-th* edge of the *i-th* cell. Considering that each $k$ RP (along the direction perpendicular to the *k-th* edge) is used to deliver information to a pair of neighboring cells of different size, the associated distance between *i-th* and *j-th* cell, $\min(l_i, L_j)$ should affect the time-step selection. Accordingly, at this point, the procedure based on RP, in the case of triangular elements, requires that the time-step size is limited by:

$$\Delta t \leq CFL \cdot \frac{\min(l_i, l_k)}{\max\limits_{m=1,2,3} \left| \widetilde{\lambda}_m \right|} \qquad (11)$$

where *CFL* is *Courant-Friedrichs-Lewy* number (for 1D case $CFL = (u\Delta t)/\Delta x$, $u$ is the actual flow velocity, $\Delta t$ the numerical *time-step* and $\Delta x$ is the *mesh-size*) to be defined by the user (for all simulations $CFL = 1$), $\widetilde{\lambda}_m$ is a velocity of the flow perpendicular to the *m-th* edge, emerging from the application of the 1D RP solver along the three directions in the case of triangular elements. Accordingly, the selected software computes automatically the time step that results to be variable during the transitory, proportional to the local grid-cell width, but also inversely proportional to velocity and depth (Equation (11)).

### 2.5. Conceptual Model and Selected Inputs

The topographic data set was based on a $5 \times 5$ m grid resolution DTM and on a 1:10,000 scale Regional Technical Chart obtained during the years 2009–2011, both available at on-line Geo-site of the Piemonte Regione, [40] WGS84, UTM Zone32N. The boundary of

the geometric domain was traced considering the extension of the water catchment area of the Marderello torrent, embedding the hydrometer placed downstream of the waterfall. Free Inflow and Free-Outflow conditions were imposed, respectively, at upstream and downstream border of the water catchment area. The geometric domain of the conceptual model embedded the whole contours of the river basin.

No water sources were assumed in particular along the lateral borders and, accordingly, no lateral boundary flow was assumed. The beginning of the numeric transient (22 July 2016, 5:40) was set long after the previous rainy event, which in any case did not generate any flooding phenomenon, as it may be inferred from the inspection of Figure 2. Moreover, the terrain under study was assumed in initial dry conditions.

### 2.5.1. Numerical Meshing and Spatial Rainfall Distribution

The numerical tessellation of the selected DTM was performed through unstructured mesh approach, based on linear triangular elements, applying the meshing tool accompanying the Riverflow2D computer code. Figure 3a,b (freely adapted from [9]), display the numerical discretization of the spatial domain, selected to apply the methodology under consideration and, in agreement with the purpose of the paper, to meet both the requirement for numerical accuracy (high number of tessellation elements) and the request of very fast running time. Therefore, we tried to optimize the distribution of the number of elements, more numerous where the morphology required a better detail. First simulations were based on different uniform average grid-cells width in order to calibrate the model against the mesh refinement. Numerical domain tessellations, performed using triangles with an average size of 10, 5 and 2.5 m were tested. The 10 m grid-cells highlighted the presence of small, rather deep depression in the mid-valley downstream of the Marderello torrent, which hinders the natural flow of water.

By a grid-cells refinement, lowering their average width from 5 to 2.5 m, the depth of these numerical depressions decreased, suggesting that their occurrences were emphasized by a discretization not sufficiently accurate to correctly reproduce the topography of the area. Accordingly, in order to perform parametric study with a reasonable number of cells, four different zones were identified (Figure 3d): an upstream area (circle 1), characterized by a rather homogeneous surface, with 10 m triangles average width (TAW); a central area (circle 2), characterized by a homogeneous surface, but with deeper incisions than the previous one, with a 5 m TAW; a downstream area (circle 3), characterized by depressions, with 2.5 m TAW and, within this area (circle 3), a further subdivision delimiting the riverbed of the Marderello torrent, characterized by 2 m TAW (circle 4). Then, a factor F was introduced in order to modify and identify the selected average mesh-element width described in Figure 3d. This factor describes the adopted spatial discretization of the basin. For example, F = 0.5 means that the average dimensions of the numerical mesh, which cover the 4 zones of division of the spatial domain (Figure 3d), were multiplied by a factor 2. Consequently, the 4 zones were discretized according to the average dimension values of the triangles mesh equal to, respectively, 20, 10, 5 and 4 m, instead of 10, 5, 2.5 and 2 m. The purpose of this approach was to acquire more insights on how much the results were sensitive to mesh refinement. Rainfall time histories were acquired through 5 gauges placed within the Marderello basin and, consequently, only 5 points were monitoring. These pointwise information, the only data available since no radar surveys were performed, had to be processed in some way in order to obtain an appropriate areal rainfall input. Converting point precipitation observed at a ground gauge into areal precipitation averaged over a spatial domain and estimating the ratio between them (the Areal Reduction Factor, ARF) has been extensively studied with respect to the design of hydraulic and hydrologic infrastructures for the last several decades (among many others: [41–43]).

However, in this paper, a simplified approach was selected, as will be detailed in the following sections. The occurrence of windy phenomena that would affect only the momentum components was excluded since it did not appear evident from the video taken

by the camera. Moreover, in situ inspection carried out during the period of the event under study, excluded residual snow deposits.

### 2.5.2. Rainfall Inputs

First of all, only the totality (100%) of the rain intensity measured at the Marderello station was considered, to begin exploring the sensitivity of the results from the precipitation distribution.

Simulations were then carried out by dividing the territory under study into three different main areas, as shown in Figure 3e, located at different altitudes and characterized by different rain intensities. Only point rain inputs were available and not a spatially continuous distribution as in the case of radar measurements. Accordingly, in order to consider even roughly the heterogeneity of the distribution of rainfall, different percentages of the experimental values, decreasing with the altitude of each measurement station, were assumed as average values within each area. A summary description of the resulting rainfall inputs, selected for the simulations is provided below:

- Zone 1: an upstream area with rainfall inputs equal to the 100% of the measured data obtained from the Marderello station;
- Zone 2: a central area with input equal to 60–80% of the experimental data (for details see the caption of Figure 4d–f);
- Zone 3: the downstream area equal to 30–65% of the experimental data (caption of Figure 4d–f).

Given the high rainfall, high air humidity and relatively low temperatures, for this case study, the evapotranspiration phenomena was considered negligible.

### 2.5.3. Rainfall Inputs Including 5 Available Rainfall Gauges Data

As a further and more realistic approach, all the rainfall time histories measured at the 5 available gauges were considered. In order to convert the related gauge-point data into averaged areal precipitation, the Marderello basin domain was divided into 5 polygons based, at first, on Thiessen's (also called Voronoi's polygonization) tessellation (Figure A2). Then, the rainfall-polygons were adapted to the already performed polygons, characterized by different width mesh in different areas of the basin, Figure 3e–g.

### 2.5.4. Computer Hardware Specifications

Some details of the computer used to perform the simulations discussed in this paper are given:

- CPU Intel i7-7700 3.6 GHz;
- #Cores: 4;
- #Threads: 8;
- GPU Nvidia GeForce GTX 1060 6 GB;
- #GPU Cores: 1280;
- Clock: 1506 MHz;
- Ram: 32 Gb.

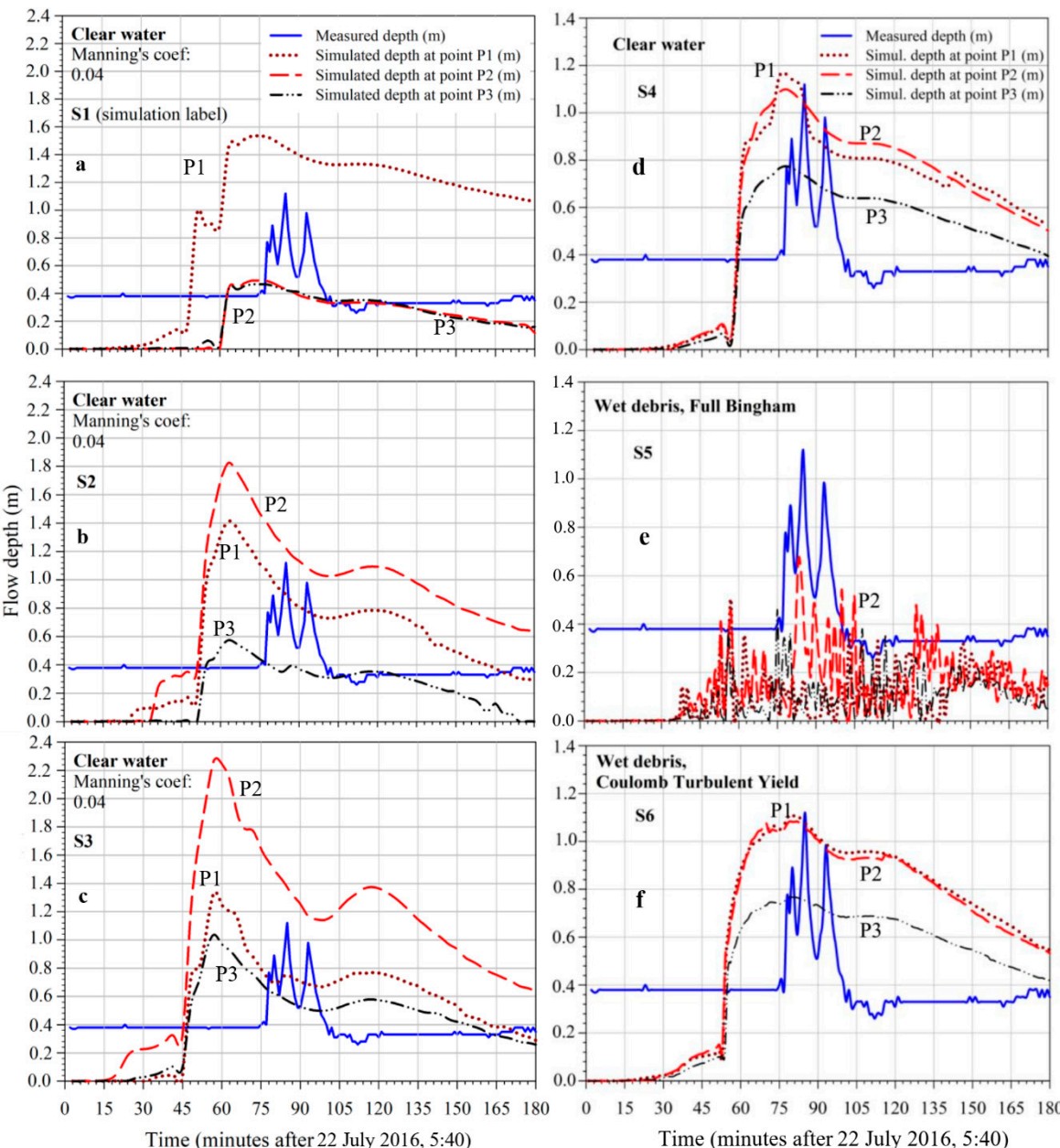

**Figure 4.** First attempts, clear-water, uniform mesh and uniform rainfall distributions: (**a–c**); First attempts, clear-water and mud-models options, variable mesh, variable rainfall distribution: (**d–f**). (**a**) Simulation S1: u.a.m.w. = 10 m; (**b**) S2: u.a.m.w. = 5 m; (**c**) S3: u.a.m.w. = 2.5 m; (**d**) S4, Clear-water, Rainfall distribution *VarRainDistr-1*, Manning coefficient = 0.05, viscosity = 0.01 (Pa·s); (**e**) S5, Wet-debris, Full Bingham (FB), Rainfall distribution *VarRainDistr-2*, Manning coefficient = 0.055, Yield tension 0.075 (Pa); (**f**) S6, Wet-debris, Rainfall distribution *VarRainDistr-3*, Manning coefficient = 0.055, CTY Yield tension 0.075 (Pa); Angle = 3°; (**d**) S7, Wet-debris, Rainfall distribution *VarRainDistr-4*, Manning coefficient = 0.055, CTY Yield tension 0.075 (Pa), Angle = 3°.

## 3. Results and Discussion

Many simulations were performed in order to calibrate both the selected type of flow and the values of the parameters characterizing the rheology. To this purpose, we selected the comparison of the profiles of the muddy wave over time, resulting from numerical simulations, versus the data measured at available hydrometer. Many issues and topics were included for the numerical parametric analyses: width and type of distribution of the triangles that covered the spatial domain of the basin, rainfall distribution and selected gauge stations, values of the Manning's coefficient and of other important rheological

parameters, inclusion or exclusion of infiltration phenomena. The simulations were divided into two main different groups. For the first group of simulations, as first attempts described in Table 1 (simulations S1–S7), only the Marderello gauge was considered as rainfall input, while in the second group, Table 2 (simulations S8–S30), all the rainfall contributions, measured at the five gauges, were included. Moreover, a further subdivision was the selection between clear or muddy water models. We identified the maximum value of the wave crests and the wave arrival time as the two most important physical characteristics against which to perform and develop the parametric studies and comparison.

**Table 1.** Summary description of the main parameter numerical values assumed for preliminary simulations based on only Marderello rainfall time history; TMR: Total Measured Rainfall (mm/h); *VarRainDistr-1, -2, -3, -4* variable rainfall distributions, percentage (%) of TMR, related to the selected areas 1, 2, 3; MD: Mud-Debris module; FB: Full Bingham; CTY: Coulomb-Turbulent-Yield; u.a.m.w.: uniform average mesh width (m), uniform (*Unif.*) within the whole area or uniform (*VarMeshDistr*) but with a different value in each of the 1, 2, 3, 4 subdomains areas.

| Simul. (Figure) | Rainfall Distribution Figure 2c Marderello Gauge Percentage (%) of TMR | Debris (MD Module) | Mesh (m) | Manning (n) | Density (kg/m³) | Yield stress (YS) $\tau_{yield}$ (Pa) | Viscosity (Pa·s) | Angle (°) |
|---|---|---|---|---|---|---|---|---|
| S1 Figure 4a | Uniform (100%) | no | *Unif.* 10. u.a.m.w. | 0.04 | 1000 | - | 0.01 | - |
| S2 Figure 4b | Uniform (100%) | no | *Unif.* 5 u.a.m.w. | 0.04 | 1000 | - | 0.01 | - |
| S3 Figure 4c | Uniform (100%) | no | *Unif.* 2.5 u.a.m.w. | 0.04 | 1000 | - | 0.01 | - |
| S4 Figure 4d | Figure 3e *VarRainDistr-1:* areas ① ② ③ 100−60−30% | no | Figure 3d *VarMeshDistr-1:* areas ① ② ③ ④ u.a.m.w.: 10 m 5 m 2.5 m 2 m | 0.05 | 1000 | - | 0.01 | - |
| S5 Figure 4e | Figure 3e *VarRainDistr-2:* 100−80−62% | FB | Figure 3d *VarMeshDistr-1* | 0.055 | 1100 | 0.075 | 0.1 | - |
| S6 Figure 4f | Figure 3e *VarRainDistr-3:* 100−80−64% | CTY | Figure 3d *VarMeshDistr-1* | 0.055 | 1100 | 0.075 | - | 3 |
| S7 Figure 4g | Figure 3e *VarRainDistr-4:* 100−80−66% | CTY | Figure 3d *VarMeshDistr-1* | 0.055 | 1100 | 0.075 | - | 3 |

### 3.1. First Attempts with Marderello's Gauge Rainfall Inputs

In Table 1 details are given of the simulations carried out assuming as rainfall inputs only the Marderello's gauge contribution. In order to gain some feeling on how much a debris-flow modelling approach could affect the results, the first simulations were performed considering just clear water without the inclusion of the MD (Mud-Debris) module.

A total of three points belonging to the numerical model, very close to each other, were taken as references: P1 upstream point, P2 located at the actual experimental measurement point, P3 downstream point (Figure 3a–c). For the first three simulations, uniform mesh discretization was adopted (Figure 4a–c freely adapted from [9] and Table 1), applying a progressive mesh refinement through a decrease in the average dimension of the triangular elements. Then, we checked (Figure 4d–f, freely adapted from [9]) how the mesh width refinement affected the total mass flowing out the Marderello's basin through the basin's outflow boundary of the region 3, indicated in Figure 3d. With this aim we performed parametric analyses, varying the number and the average width of the triangular elements. By inspection of the resulting plot (not reported), based on the first simulation, it appeared

that the asymptotic value of the maximum height of the water level was almost reached employing about more than 2 million of 1 m average width elements. Additionally, the rainfall distribution was considered as uniform and equal to 100% of the measured data. The numerical value of the Manning's coefficient was set equal to 0.04. For all *clear-water* simulations, a water density of 1000 kg/m$^3$ was adopted. In Figure 4 the comparison between the flow depth resulting from numerical simulations at the 3 selected proxy points (see also Figure 3) and the measured data are reported. For S4, variable mesh and rainfall spatial distribution were adopted, whose details are given in the scheme reported in Figure 3d–g and in Table 1. For the S5, S6 and S7 simulations, Full Bingham (FB) and Coulomb Turbulent Yield (CTY) *mud-models*, were considered, respectively. An important issue was the identification of the mud-water density for the event under study that typically [25] amounted to around 2000 kg/m$^3$.

This value was measured at the end of the process, when and where the rainwater coming from upstream was completely transformed into mud. However, the mathematical model we selected does not allow us to consider the time and spatial variability of numerical values of some important physical parameters like the density of the muddy flow, the internal viscosity, the yield stress, as highlighted previously. Accordingly, for these reasons, in order to consider an averaged value that was able to take into account the water path, water density values ranging from 1100 (just for S4) to 1600–2000 kg/m$^3$ were explored.

*3.2. Numerical Outcomes Resulting from the Inclusion of the 5 Available Rainfall Gauges*

In Table 2, the numerical values of important parameters selected for simulations S8–S30, are reported. All the available five rainfall gauges were considered. The mandatory conversion of their pointwise data into areal precipitation over the Marderello's basin was carried out through two strategies: the first one was to consider just the Thiessen's tessellation (Figure A2; simulations S8–S18, displayed in Figure 5), while the last one was an adaptation of the Thiessen's meshing, considering also the different altitudes of each area of the domain (Figure 3f,g; simulations S19–S25 shown in Figure 6 and S26–S30 in Figure 7). The selected debris density values ranged from 1600 to 2000 kg/m$^3$. For simulations S8–S28, the *Coulomb-Turbulent-Yield* model for the mud debris flow was assumed, while the *Turbulent* option was explored for simulations S29–S30. The *rheological friction angle* was set equal to 3°, except for simulations S26 and 28, for which the angle was selected to be equal to 3.5°, while the *Turb* option did not require any value. Then, the Horton's infiltration option was explored, varying the related parameters (simulations S13–S30). For all simulations, the meshing step was based on an initial random distribution of the points that were automatically linked together through a second step based on usual approach like Delaunay's triangular tessellation. Accordingly, it was reasonable to presume that some differences could occur among outcomes obtained through different runs based on the same initial conditions and the same meshing procedure.

Hence, in Figure 6, simulations S19–S21 were compared to their repeated runs S19R–S24R (R stands for "repeated").

The simulations, whose outcomes were reported in Figure 7, were characterized by the Manning' s coefficient equal to 0.15, two times the previous attempts. The inputs difference between S27 and S28 consisted essentially of different adopted values of filtration parameters, as reported in Table 2, while for the simulations S29 and S30 the *Turb* rheological option was explored, with different infiltration parameters.

Finally, in order to verify the important computing time reduction requirement, mandatory for the aims of possible early-warning system, we performed some of the discussed simulations through two different modalities featured by the code, as it will be discussed in the next paragraph.

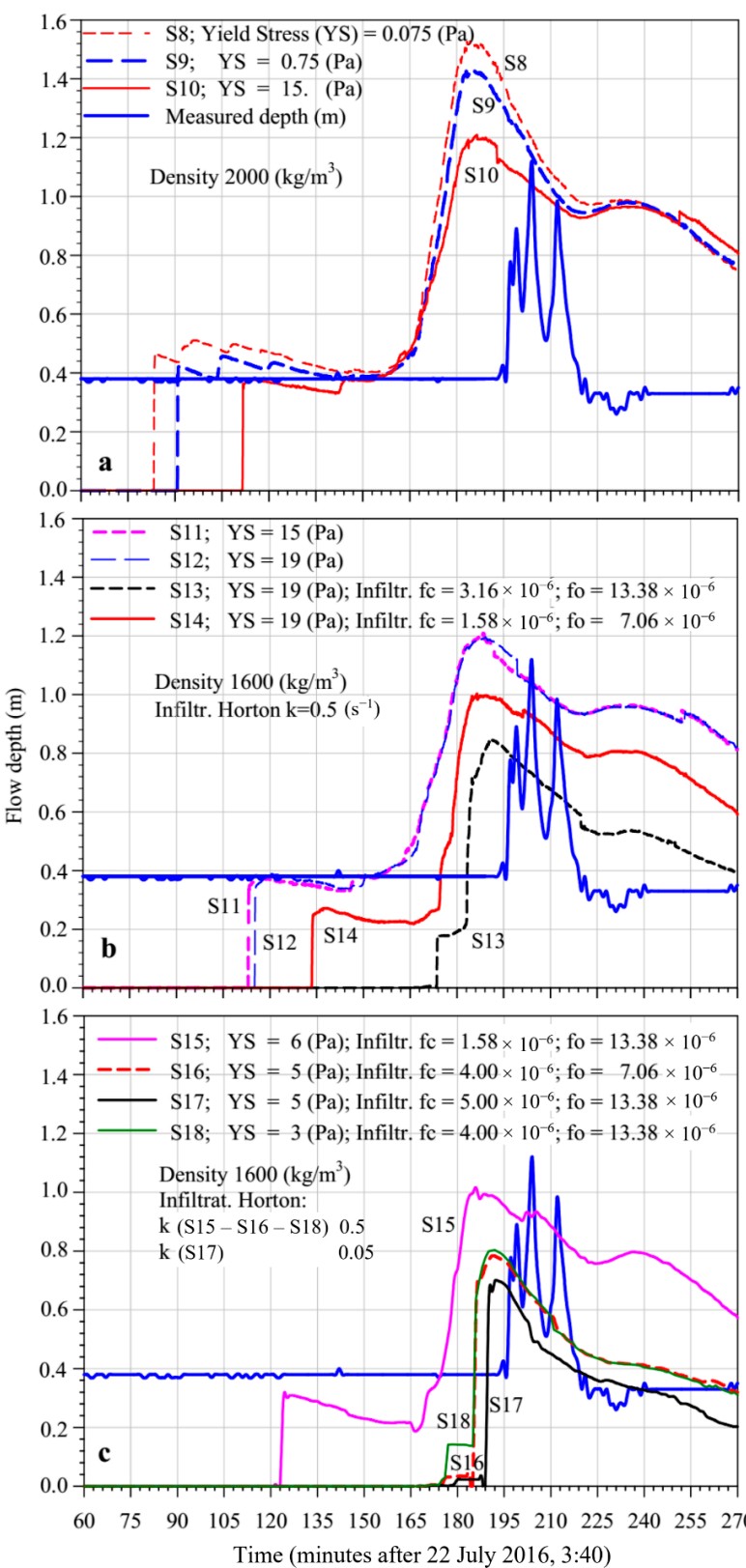

**Figure 5.** Inclusion of the available 5 rainfall time histories; Manning's *n* factor = 0.055; Angle = 3°; (**a**) no infiltration, Yield stress variations, density equal to 2000 kg/m$^3$; (**b**) Horton's infiltration model, Yield stress variation, infiltration parameters variation, *k* = 0.5 s$^{-1}$, density equal to 1600 kg/m$^3$; (**c**) Horton's infiltration model, Yield stress variation, infiltration parameters variation, *k* = 0.5 s$^{-1}$ and *k* = 0.05 s$^{-1}$ for simulation S17, density equal to 1600 kg/m$^3$.

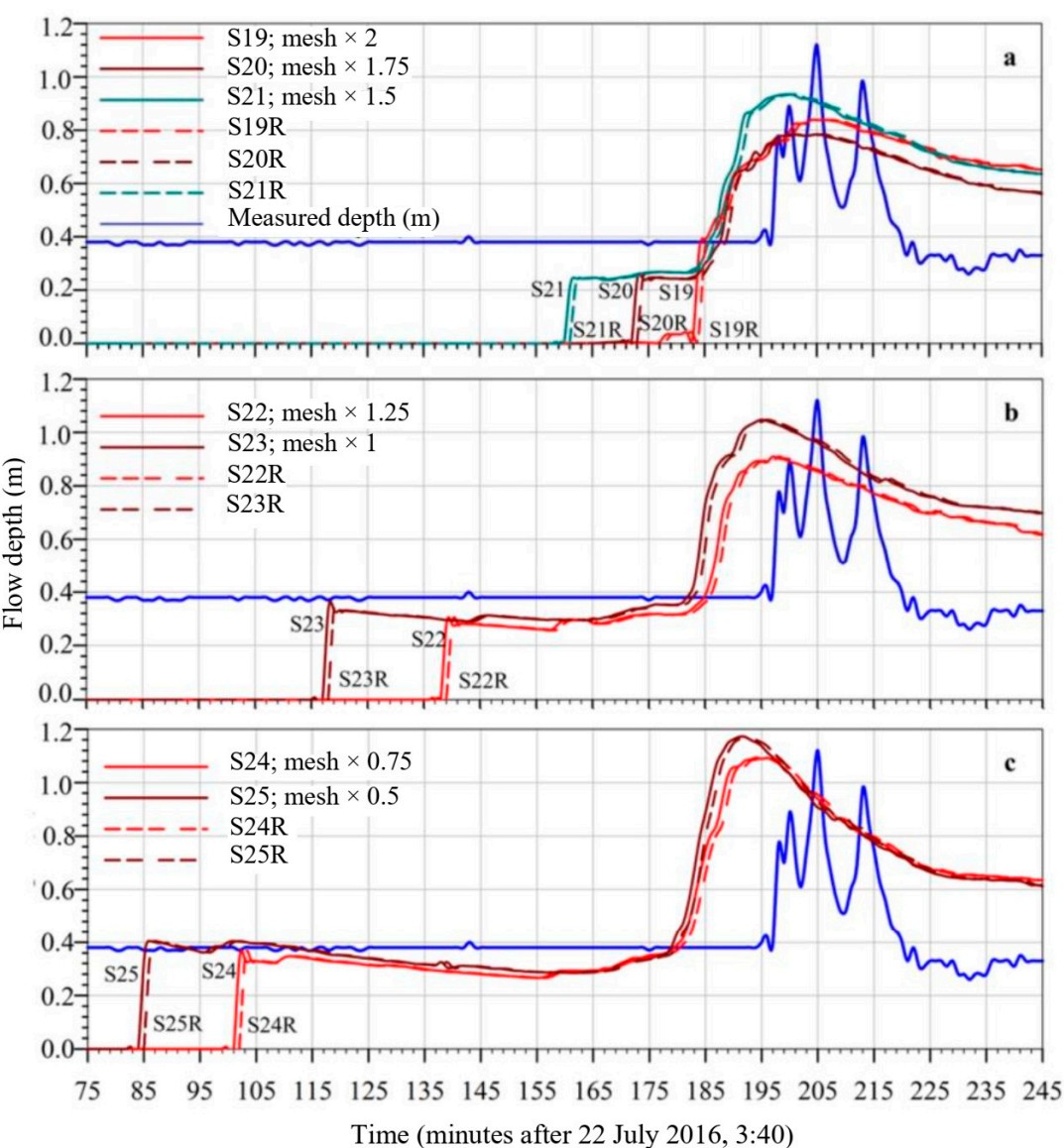

**Figure 6.** Analysis of the sensitivity of the numerical results to the average width of the triangular elements; Density = 1600 (kg/m$^3$); Manning's coefficient = 0.055, CTY rheological model; YS = 5 (Pa); Angle 3°; $k$ = 0.05 (s$^{-1}$); $f_c$ = 1.58 (m/s); $f_0$ = 7.06 × 10$^{-6}$ (m/s); (**a**) comparison of simulations S19–S21 to their repeated runs S19R-S21R; (**b**) comparison of simulations S22–S23 to their repeated runs S22R–S23R; (**c**) comparison of simulations S24–S25 to their repeated runs S24R–S25R.

### 3.3. Simulations Comparison

As initial step, just one rainfall time history, measured at the gauge located at the Marderello's station (Figures 1 and 3a,b) was considered. From Figure 4a–c and simulations S1–S3, it emerged that the refinement of the uniformly distributed mesh width (from 10 m to 2.5 m average values) implies an increase in the water level resulting from the numerical simulations. This occurrence may be due to an increase in the total cumulative rainfall inventory or to an increase in the easiness with which water could flow through a topography rich in incisions, wells and sinks that can virtually trap numerically the water if the meshing is not adequate. This circumstance was observed also on the phase of the morphological calibration of the number of cells, as observed in Section 2.5.1. In any event, this issue needs to be further analyzed. From the same figure, it clearly appears that the *FlowArrTime* (time at which the computed flow height increases very quickly from 0.4 m value) and the *PeakArrTime* (computed time of the maximum peak occurrence)

anticipated, respectively, about 30 and 25 min the values actually measured, respectively, at the 77th and the 84th minute after 5:40, 22 July 2016, the difference in time being about 36 min between the occurrence of the highest peak of the rain and the arrival of the measured maximum peak of the debris flow. Furthermore, the *ResFlowBeforePeak* (residual flow-depth shape, around 0.4 m, before peaks occurrence) was almost not foreseen by the numerical simulation, while the *ResFlowAfterPeak*, (residual flow-depth after peaks occurrence) notwithstanding a second numerical surge emerged from the calculation, was overwhelming with respect to the measured shape, as for all the simulations performed in this paper. Notwithstanding our use of the GPU parallel approach option of the code, the computational time was too high to perform parametric analyses in a reasonable time. Moreover, in order to explore the capability of the model to be used for early-warning purposes, the computational time should be as short as possible. Furthermore, to lower the number of necessary mesh elements to cover the spatial domain, we selected four areas in which different numbers of cells were imposed, as detailed in Figure 3d,g. On the other hand, for these first simulations, the infiltration was not considered. The results are reported in Figure 4d–f. Simulation S4 showed that, notwithstanding the *clear-water* assumption was again adopted, a significant improvement over previous simulations, based just on uniform rain distributions and water inventory, was obtained through a reasonable calibration of the rainfall distribution within the basin areas. Specifically, in effect, the anticipation of the *FlowArrTime* was reduced. Moreover, the *HighPeakVal* (*HPV*) (calculated highest debris flow height) fit the measured values quite well. Then, we explored the results obtained applying FB and CTY rheological models. The FB's model and assumption selected to perform S5, proved to be capable at predicting the occurrence of multiple surges, due to "stop and go" mechanisms, according to the requirement that the material motion may occur only if a threshold of the yield stress is overcome, specific to this kind of phenomenology. However, due to uncertainties related to morphology and to other parameters, this rheological model may be not completely reliable for the early-warning purposes and at this not-very-detailed scale of the phenomena. Then, the CTY approach was explored to perform both S6 and S7 (Figure A3) simulations that differ just for a slight amount in the rainfall inventory within the area in which the hydrometer was located. However, the difference was also due to the different meshing obtained automatically by SMS for the two simulations, as will be discussed in the following. The outcomes from S6 and S7 predicted an increase in the anticipation of the *FlowArrTime* (*FAT*) and a "widening" of the flow depth curve compared to the outcomes from the S4 simulation, based on a different rainfall distribution (Table 1) and, accordingly, on a lower mass inventory. In particular for this first set of simulations, it should be highlighted that the assigned values of the Manning's coefficient, the Yield stress, the density and the internal viscosity, notwithstanding founded on as possible as physically based considerations, were employed as calibration parameters in order to fit the measured data.

The second set of simulations (S8–S30, Table 2), based on the rainfall time histories measured at the available five gauges, provides a more realistic modelling of the phenomenology under study.

The comparison among the simulations S8, S9 and S10 (Figure 5a) shows that a 200-fold increase in yield stress, from 0.075 Pa to a more realistic value of 15 Pa, leads to a lowering of the flow height peak, respectively, from 1.55 to 1.21 m, equal to a multiplicative factor of just about 0.78, considering density of the fluid equal to $2000 \, \text{kg/m}^3$. It is important to note that the *FAT,* the *PAT* and the *RFAP*, were not affected by the selected value of the yield stress. Comparing to the previous simulations, the *RFBP* of the simulations reported in Figure 5a corresponds much more to the measured trend than the previous simulations. Rainwater, falling and flowing, eroded the soil and the density of the flowing material increased. Therefore, the density of the debris increased from a value of $1000 \, \text{kg/m}^3$, to a value around $2000 \, \text{kg/m}^3$, within the basin (Table 2). However, the selected computer code does not allow us to include density variation. Hence, an average value of $1600 \, \text{kg/m}^3$, more realistic than $1000 \, \text{kg/m}^3$, was adopted. Nevertheless, the comparison between

S10 (Figure 5a) and S11 (Figure 5b) shows no appreciable difference. Furthermore, an increase in the YS from 15 Pa (S11, Figure 5b) to 19 Pa (S12, Figure 5b) did not affect the results. From S13 (Figure 5b) on, the Horton's infiltration model was exploited. As expected, and as Figure 5b and c show, infiltration affected significantly the numerical results, in particular lowering the *HighPeakVal* as a consequence of the muddy water inventory lowering. The *ResFlowAfterPeak* improved considerably, as does the *FlowArrTime*, even if not so much the *ResFlowBeforePeak*. From the comparisons between S14, S15 and between S16, S18 (Figure 5b,c), taking into account the previous considerations, according to which for observing even slight variations of *HPV* the yield stress must vary greatly (S8, S9, S10, Figure 5a), it appears evident and, certainly, not surprising that the most conditioning parameter of the muddy water inventory was the $f_c$ (m/s) coefficient that is the final infiltration capacity (Equation (7)). An increase of 153% of $f_c$ led a 38% drop of the *HighPeakVal*. It is very interesting to note that increasing the $f_c$ coefficient (S17, Figure 5c) the shape of the arrival wave becomes increasingly steep, corresponding more and more to the actual sudden arrival of the mud wave. In the next step, on the basis of a plausible set of the main parameter values, affecting the outcomes of the applied CTY rheological model and already identified through the previous discussion, different simulations were conducted in order to test the results sensitivities to the variation of the average mesh width (Figure 6 and its caption). Moreover, it was analyzed the influence on the numerical results of the particular method adopted by the meshing preprocessor SMS of the selected Riverflow2D code, based on initial random determination of nodes and in the subsequent construction of the mesh through Tiessen's method. From the comparisons between S19–S25 and the correspondent S19R–S25R simulations (Figure 6a–c), it clearly appears that different initial random distributions of the nodes impacted very slight on the results. Furthermore, the mesh refinements, enhanced the *HighPeakVal*, anticipated the *FlowArrTime* and improved considerably the *ResFlowBeforePeak*. However, no substantial improvement can be observed concerning *ResFlowAfterPeak*. For the last set of simulations, S26–S30 reported in Figure 7a,b, a more realistic value equal to 0.15 of the Manning's coefficient was considered, based on an average mesh size (sub-Section 2.5) characterized by F = 0.75 (S27, S29, S30) and F = 0.5 (S26, S28). S26 and S27 differ for only the factor F. As experienced for simulations reported in Figure 6, the mesh refinement involves a higher *HighPeakVal* and an advance of the numerical occurrence of the *ResFlowBeforePeak*. However, unlike the similar trend of the curves resulting from S24 (F = 0.75) and S25 (F = 0.5), the *ResFlowAfterPeak* trends for the two S27 (F = 0.75) and S26 (F = 0.5) simulations were different. These four simulations were characterized by the same infiltration coefficient, but by a different Manning's coefficient and Angle. S24 and S27 differ just for the Manning's coefficient and the YS values (Table 2), with the same angle. As verified above, the YS value variation within the range of 5–19 Pa does not significantly affect the results. However, as Figures 6b and 7a show, there is a clear difference of the trends of the two curves resulting from S24 and S27. Accordingly, the Manning's coefficient increase from 0.055 to 0.15, affected significantly all the comparison index, in particular the *FlowArrTime*, as it could be expected since an increase in the Manning coefficient, in some way related to the interaction of the flow with the bed, implies a delay of the arrival of the flood wave. In Figure 7b, S30 simulation seems to be quite satisfactory. From Figure 7, it appears that the simulation S28, compared to measured data, provides a good compromise concerning the numerical profile of the height of the debris flow before the arrival of the peak, the arrival time of the wave, the value of the maximum wave height and the coincidence of the arrival times of the peak itself. Accordingly, in Figure 8, the comparisons of the numerical curves obtained at the three selected stations P1, P2, P3 to the rainfall data and the measured hydrometric data are given. It is important and useful to observe that an estimate of the hydrological time of concentration may be acquired by the inspection of figures similar to Figure 8, observing the lag-time between rainfall and the numerical discharge peaks, if experimental data are not available.

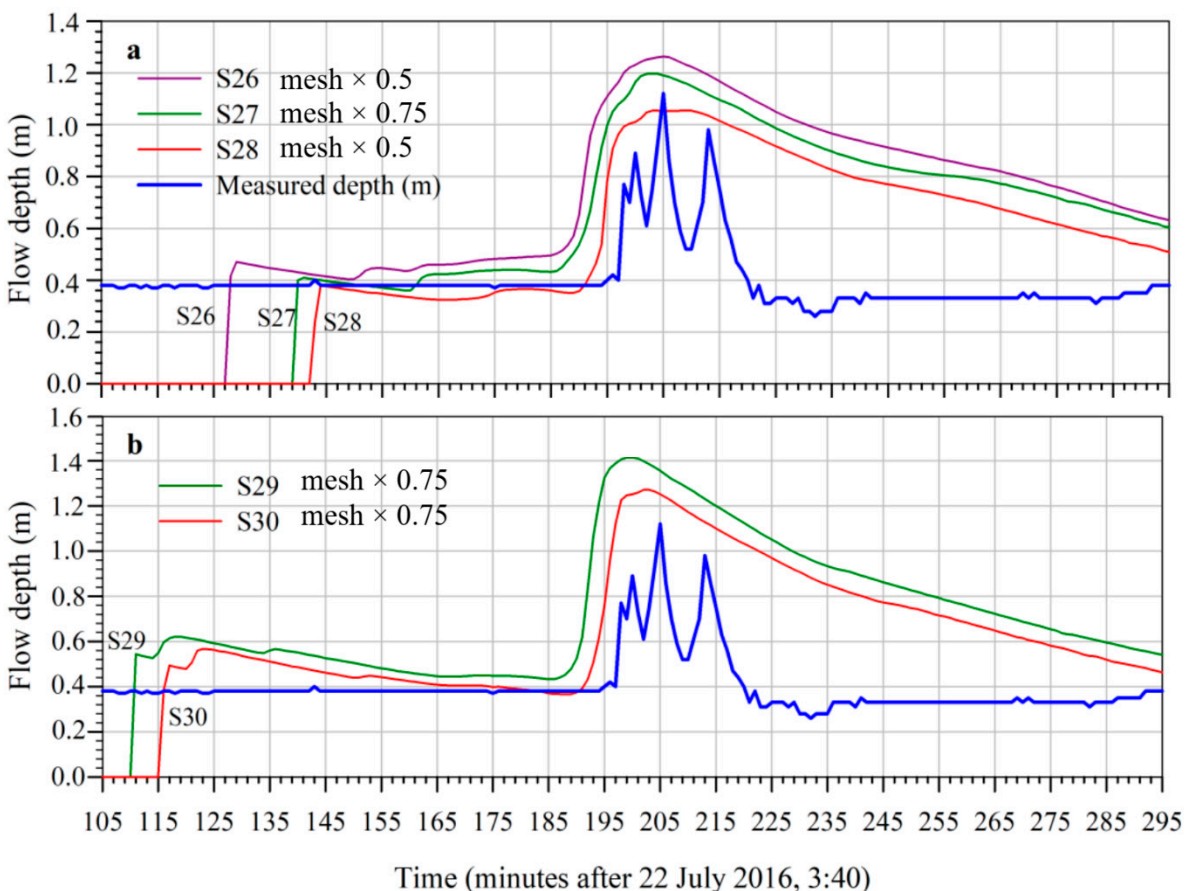

**Figure 7.** Simulations S26–S30 based on the Manning's value n equal to 0.15; (**a**) S26–S27–S28 exploring the sensitivity of the CTY modelling outcomes to grid-cell width and infiltration parameters values; (**b**) S29–S30 exploring the sensitivity of the *Turb* modelling outcomes to only infiltration parameters values.

In Figure 9a, comparison between the outcomes resulting from simulation S24, the measured flow depth and the outflow from the basin are reported (basin's outflow boundary of the region 3, Figure 3d). The time difference between the two peaks was about 20 min.

Moreover, in Figure 9b, an interesting overview of the trend of the time-steps during the transient and of the total calculation time in both CPU and GPU modes are given for simulation S24, taken as an example, but the related considerations are valid for all other simulations.

The considerable drop of the time-step during the final part of the transient was due to the circumstance that the muddy reached the areas with smaller cell-sizes. This information is also important for optimizing the grid cell size aimed at lowering calculation time in order to design an early warning system. Then, from the inspection of the resulting graphics, we verified that the outcomes were perfectly similar (except the slight differences due to the random meshing, as discussed in the following) and that the computing time decreased by 3–4 times. On the other hand, it is now commonly accepted internationally that the GPU technique, compared to the CPU approach, can decrease the calculation time by a factor that varies from 20 to 80, even using inexpensive laptops or desktops, depending on the selected hardware and software (among many others: [44]).

In order to introduce a "fitting-rank" related to simulations characterized by the same rheological and infiltration parameters values, but by different mesh size marked by the factor F, simulations group labelled S19–S30 were selected.

**Table 2.** Summary description of the main numerical parametric simulations resulting from the inclusion of the 5 rainfall gauges and also the Horton's infiltration model; F: factor that multiples the average cell width; Turb: Turbulent; Fo and Fc: the initial and final infiltration capacities (m/s); k: the rate of decrease in the capacity (s$^{-1}$).

| Simul. Figure | Mesh (m) Figure 5a | Rainfall (mm/h) All Gauges Figure 2a–e | Density (kg/m$^3$) | Manning (n) | Debris (MD Module) | YS $\tau_{yield}$(Pa) | Angle (°) | Infiltration (Horton) | | |
|---|---|---|---|---|---|---|---|---|---|---|
| | | | | | | | | k (s$^{-1}$) | Fc (10$^{-6}$) (m/s) | Fo (10$^{-6}$) (m/s) |
| S8, Figure 5a | F = 1.00 | Figure 6a | 2000 | 0.055 | CTY | 0.075 | 3 | - | - | |
| S9, Figure 5a | F = 1.00 | Figure 6a | 2000 | 0.055 | CTY | 0.75 | 3 | - | - | - |
| S10, Figure 5a | F = 1.00 | Figure 6a | 2000 | 0.055 | CTY | 15 | 3 | - | - | - |
| S11, Figure 5b | F = 1.00 | Figure 6a | 1600 | 0.055 | CTY | 15 | 3 | - | - | - |
| S12, Figure 5b | F = 1.00 | Figure 6a | 1600 | 0.055 | CTY | 19 | 3 | - | - | - |
| S13, Figure 5b | F = 1.00 | Figure 6a | 1600 | 0.055 | CTY | 19 | 3 | 0.5 | 3.16 | 13.38 |
| S14, Figure 5b | F = 1.00 | Figure 6a | 1600 | 0.055 | CTY | 19 | 3 | 0.5 | 1.58 | 7.06 |
| S15, Figure 5c | F = 1.00 | Figure 6a | 1600 | 0.055 | CTY | 6 | 3 | 0.5 | 1.58 | 13.38 |
| S16, Figure 5c | F = 1.00 | Figure 6a | 1600 | 0.055 | CTY | 5 | 3 | 0.5 | 4.00 | 7.06 |
| S17, Figure 5c | F = 1.00 | Figure 6a | 1600 | 0.055 | CTY | 5 | 3 | 0.05 | 5.00 | 13.38 |
| S18, Figure 5c | F = 1.00 | Figure 6a | 1600 | 0.055 | CTY | 3 | 3 | 0.5 | 4.00 | 13.38 |
| S19, Figure 6a | F = 2.00 | Figure 6c | 1600 | 0.055 | CTY | 5 | 3 | 0.05 | 1.58 | 7.06 |
| S20, Figure 6a | F = 1.75 | Figure 6c | 1600 | 0.055 | CTY | 5 | 3 | 0.05 | 1.58 | 7.06 |
| S21, Figure 6a | F = 1.50 | Figure 6c | 1600 | 0.055 | CTY | 5 | 3 | 0.05 | 1.58 | 7.06 |
| S22, Figure 6b | F = 1.25 | Figure 6c | 1600 | 0.055 | CTY | 5 | 3 | 0.05 | 1.58 | 7.06 |
| S23, Figure 6b | F = 1.00 | Figure 6c | 1600 | 0.055 | CTY | 5 | 3 | 0.05 | 1.58 | 7.06 |
| S24, Figure 6c | F = 0.75 | Figure 6c | 1600 | 0.055 | CTY | 5 | 3 | 0.05 | 1.58 | 7.06 |
| S25, Figure 6c | F = 0.50 | Figure 6c | 1600 | 0.055 | CTY | 5 | 3 | 0.05 | 1.58 | 7.06 |
| S26, Figure 7a | F = 0.50 | Figure 6c | 1600 | 0.15 | CTY | 19 | 3.5 | 0.05 | 1.58 | 7.06 |
| S27, Figure 7a | F = 0.75 | Figure 6c | 1600 | 0.15 | CTY | 19 | 3 | 0.05 | 1.58 | 7.06 |
| S28, Figure 7a | F = 0.50 | Figure 6c | 1600 | 0.15 | CTY | 19 | 3.5 | 0.05 | 2.5 | 8.50 |
| S29, Figure 7b | F = 0.75 | Figure 6c | - | 0.15 | Turb | - | - | 0.05 | 1.65 | 7.48 |
| S30, Figure 7b | F = 0.75 | Figure 6c | - | 0.15 | Turb | - | - | 0.05 | 2.5 | 8.5 |

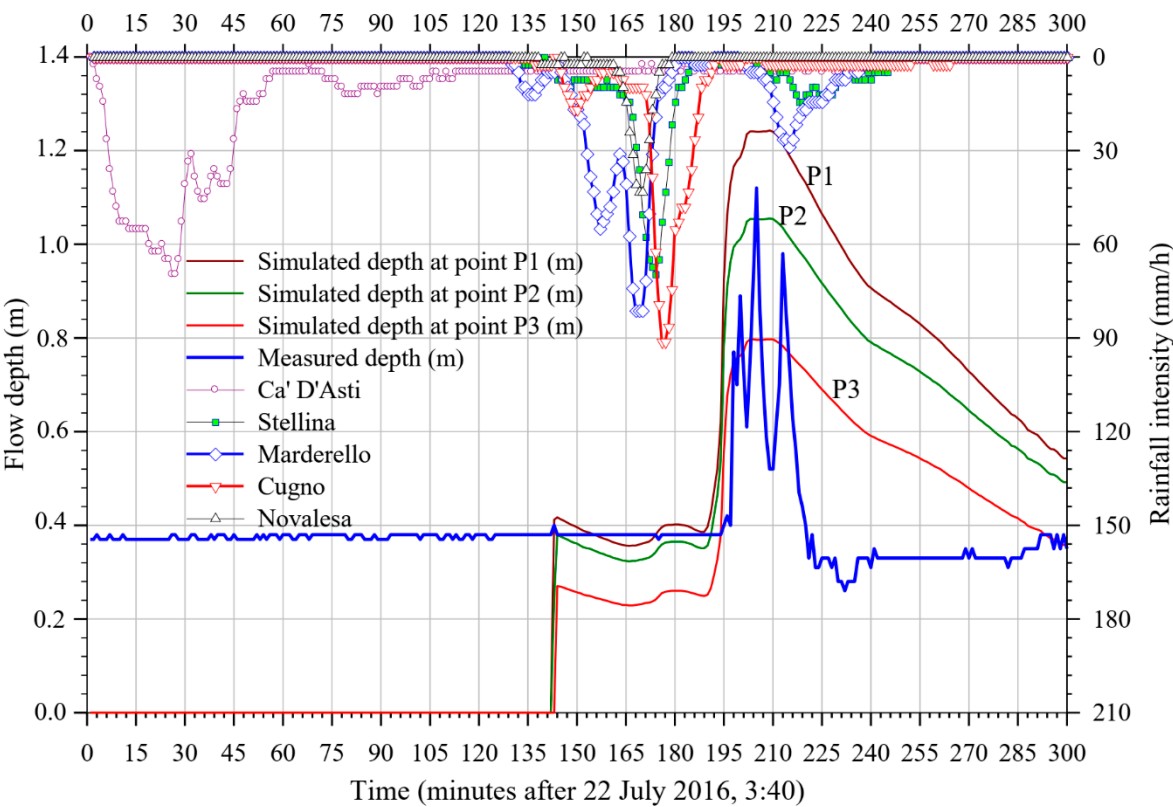

**Figure 8.** Simulation S28; comparison among the flow depth curves resulting at the 3 selected 'numerical stations' (P1, P2, P3) and the data measured at pluviometric and hydrometric stations.

Hence, in Figure 10 for each average grid-cell size identified by the F factor (Figure 10a), according to the corresponding number of nodes, the *FAT*, *HPV*, *PAT* and the computational time in GPU mode are given. Moreover, the percentage of the deviation among the numerical and the corresponding measured values were specified:

$$Percent.(\%) = \frac{Var_{numerical} - Var_{measured}}{Var_{measured}} \qquad (12)$$

where $Var_{numerical}$ and $Var_{measured}$ are the values of the selected variable, as reported in Figure 10a,b.

Figure 10a,b's result is very useful to identify, for the models selected for the simulations, the number of nodes with the relative distribution of the cell average size, above which there is no improvement in the correspondence of the calculated parameters, fundamental for early-warning system such as the *HPV* (Figure 10a), the *FAT* (Figure 10b) and the *PAT* (Figure 10b). It appears that, for the set of the selected simulations, this threshold was reached for F = 0.75, with the related distribution of grid-cell average size (7.5, 3.75, 1.875 and 1.5 m, Figure 3d). The deviation of the best numerical *HPV* was only 4% of the measured value (Figure 10a), while the lowest deviation of *FAT* was about 3%. However, the GPU computational time was about 72 min, while the time difference between the rainfall spike and the arrival of the wave was about 20 min as inferred from Figure 8. Accordingly, in this case, the calculation time should be cut down by at least 10-15 times, that is, it should be taken only 5–7 min, allowing 13–15 min to alert, by acoustic alarm as well, tourists, hikers, families on an excursion to get away, in this case from the riverbed and its neighborhood. This or an even more satisfying target could be achieved by a more powerful dedicated computer with a more powerful GPU card, like Tesla Nvidia GPUs as the newest A100 GPUs, or by selecting a less accurate, but still acceptable, discretization. Finally, it should be noted that discrepancies between measured data and computed values, obtained at the beginning of the transitory and immediately after the wave peaks, are

observed in all simulations. The main reasons are due to the limitations discussed above, such as, among others:

- rainfall data acquisition based on punctual measures;
- DTM's scales coarser than those evidently necessary to capture the many details of the debris profile, in particular, the channeling of small quantities of water following different paths from those that can be measured by the only available hydrometer;
- variability of the density and the changing of the debris' typology along its path, the actual rate of water infiltration into the soil;
- the use of a necessarily simplified CFD modelling, due to the large scale of the are under study.

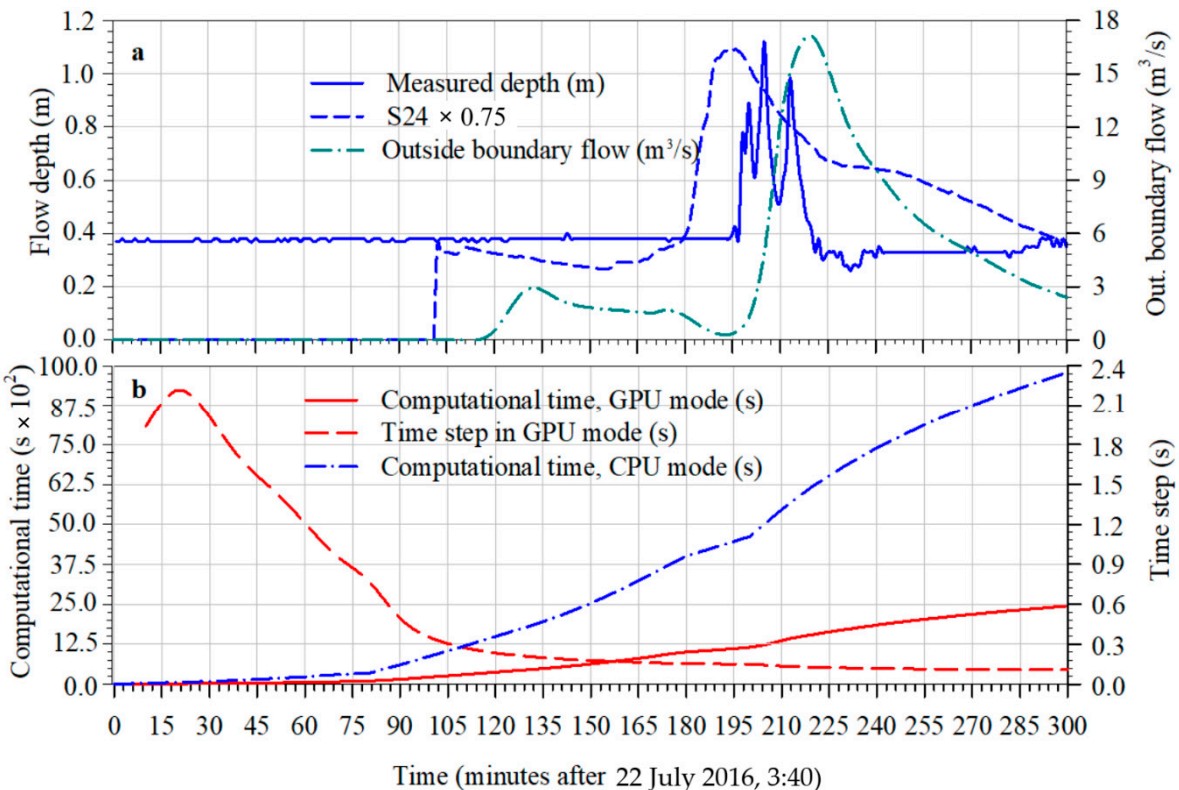

**Figure 9.** Simulation S24: (**a**) numerical flow depth at station P1 and outside-boundary water flow; (**b**) Computational time in CPU and GPU modes and GPU time step values.

Moreover, the main purpose of the simulations discussed in this paper was to predict the arrival time and the maximum peak value of the debris flow height, well captured by the selected modelling. Therefore, the prediction of multi-surges and the prediction of the wave profile before and after the arrival of the peak assumed secondary importance and beyond the target of this paper.

It is worth underlining the importance that the variability and uncertainty of the numerical values of physical, mechanical and hydrological data, as well as the uncertainties associated with the topography survey of the area under study, may assume in the elaboration of simulations of the type discussed in this article, as reported in [45,46]. Further, statistical and fractal methods could be used to analyze time series associated with rainfall in the area of interest, to identify the extent of possible flooding [47].

Finally, the plots reported in Figure 11 show the evolution over time of the water distribution that is flowing according to the numerical forecast resulting, for example, from the S15 simulation. A more detailed study of plots of this type, just outlined in this paper, could allow the optimization of the allocation of rainfall gauges within the basin under study.

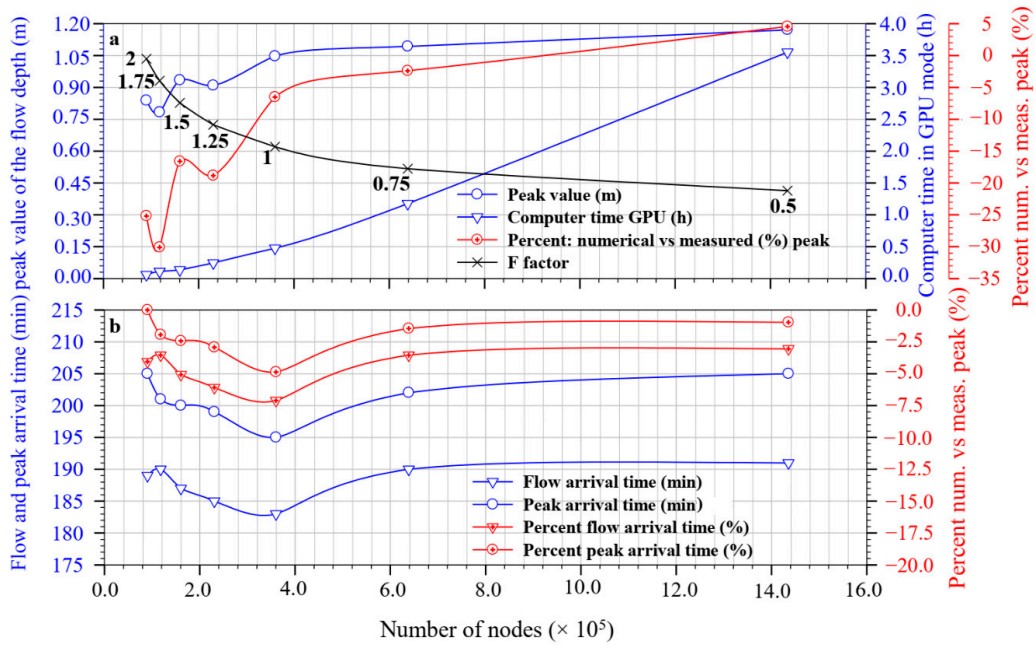

**Figure 10.** Simulations S19–S30, according to the number of nodes: (**a**) *HighPeakVal*, Computational time in GPU mode, Percentage of the deviation among the numerical values of the peak and the corresponding measured values, F Factor; (**b**) *FlowArrTime*, *PeakArrTime*, Percentage of the deviation among the numerical values of the flow and peak arrival times and the corresponding measured values.

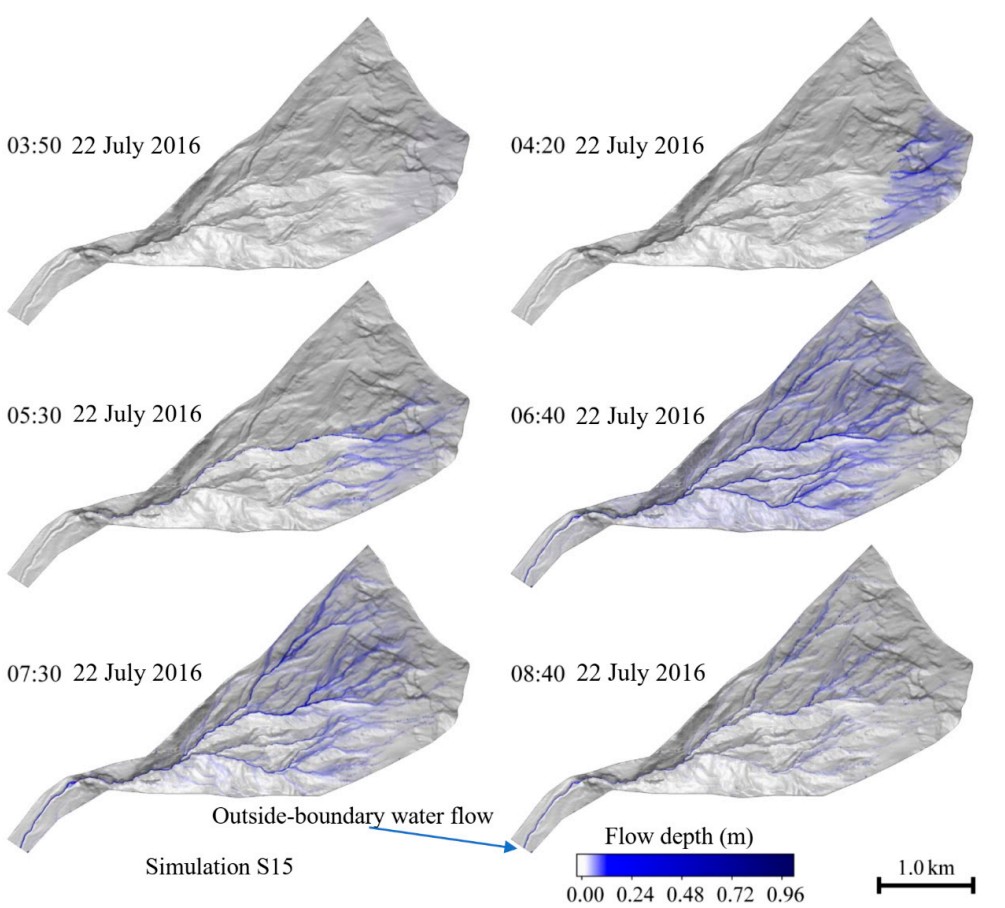

**Figure 11.** Simulation S15: visualization of the distribution of water within the basin during the debris flow occurrence transitory over time, including infiltration.

## 4. Conclusions

Notwithstanding the complexities of the geomorphology of the system and of the physics of the studied phenomena, the selected numerical approach was demonstrated to estimate well the arrival time and the peak of a real muddy debris flow phenomena (see Figures 8 and 10) that may occur within the selected test case. The proposed approach was based on DTM, pluviometric data, calibration through hydrometric measures, the Godunov *Classical Shallow Water* modelling (despite its limitations for steep slopes), the augmented approximate Riemann solvers and the GPU approach. Accordingly, the simulations proved that this kind of modelling with the necessary improvements, as discussed in the paper, could allow, quite well, the prediction of basilar features in order to develop an early warning system. The numerical outcomes are strongly affected, first of all, by a correct knowledge of the local morphologies also at small scale. In the case discussed in this paper, the intensity of erosion and the possible change of the conoid morphology during the period from 2011, the year when the only available DTM survey was performed, until 2016, the year when the studied event occurred, do not seem to have affected in a decisive way the incisions of the terrain through which the muddy debris flowed. However, after the acquisition of an available DTM and a necessary in situ inspection, a possible improvement of the initial morphological information could be mandatory, also by means of an inspection of the numerical tessellation and a consequent manual refinement. Moreover, some more important suggestions were gained in order to develop a Shallow Water model accounting for all the modelling features necessary to simulate more properly Alpine debris flow, steep slopes included. A further limitation of the applied mathematical model was that the inclusion of the variability of the physical parameters during the flow, which turned from clear water to mud, was only partially allowed. Despite this kind of simplification, the selection of available rheological laws, in particular the most promising *Coulomb-Turbulent-Yield* and *Turbulent* models, provided a satisfactory matching of the numerical estimate of the wave arrival time and the muddy flow-height to the measured data. On the other end, the outcomes of the *Full-Bingham* model reproduced the characteristic movement due to the occurrence of the plasticization tension threshold with repeated surges. This type of constitutive law requires both detailed simulation of stress distribution and accurate morphological description. Another important topic would be the acquisition of a correct rainfall distribution knowledge. The best solution would have been a radar measurement of the rain distribution, but it is not so easily available. As an alternative, this requirement would be satisfied through a correct gauges' network design of monitoring stations, for both rainfall and hydrometric measurements, that could be an important outcome deriving from this kind of study, by means of an accurate inspections of the water distribution acquired through numerical simulation, like the plot reported in Figure 11. Furthermore, an estimate of runoff times of the selected basin, may also be acquired by inspection of figures similar to Figure 8. Another issue, in our opinion not sufficiently explored and analyzed in the bibliography through proper sensitivity parameters analyses, was the infiltration of rainwater during the event under observation. Affecting the availability of runoff water, it was of considerable importance both on the arrival of the flood wave and on its height, as we have observed performing the numerical tests. The use of the GPU approach was very important (parallel computing with the Nvidia graphics card) in order to achieve the purpose of the paper, with a computational time reduction (with the available hardware) of 3–4 times of the time required by most common CPU approach. Furthermore, the selection of the GPU approach, allows to perform several simulations in an acceptable time, as well.

A more advanced mathematical models and, possibly, a more powerful hardware for GPU numerical solution, in accordance with what is discussed in this paper, would be desirable. Moreover, the simulations of more historical cases should be mandatory to validate an interdisciplinary methodology aimed to be part of a debris warning system. It is worth noting that a recent version of the RiverFlow2D, available from August 2020, introduced important changes regarding the previous version, including, for example, the

variabilities of quantities such as debris density, viscosity and yield stress. The introduction of these improvements seems to correspond quite well with some of the requirements needed to perform realistic simulations of debris flow, as emerged and discussed in this paper. However, the application of this new version will be the content of a future research.

On the other hand, the results coming from the previous discussion and the physical reasonability of the numerical-values of the geo-mechanical parameters, suitable for fine tuning the selected models, indicated that the whole methodology, although only one case was currently examined, is promising and most likely applicable to other similar events, as well.

Mathematical models based on the classical balance principles of the physical quantities of interest, such as the shallow water approach, selected for this work, certainly show greater flexibility and generalization of their use than the possible use of less complex empirical models which are still valid [9,27,28], but whose applicability is limited to the typologies of phenomena for which they have been proposed and, accordingly, cannot have a general validity. On the other hand, models based on CFD require a great amount of computing time. However, this work attempts to demonstrate that with the use of the GPU tool, sophisticated mathematical models can also be considered to develop early warning systems.

Furthermore, it is worth noting that our methodology can also be used for calibration of a classical early warning system, based on rainfall thresholds, if, of course, a consistent number of historical events is available for a specific case study.

In summary, in the specific case of Marderello and in similar debris flow events in other zones as well, in order to develop an effective early warning system, a more detailed DTM acquired by drones flights, an optimized rainwater gauges network with wireless data transmission, a very powerful graphic card based on the GPU technique, like Tesla GPU Card, suitable for reducing to just a few minutes the calculation time of a dedicated mathematical model more advanced than that implemented into the selected version of the used code, including models aimed to calculate or estimating at least the triggering formation of debris flow due to the impact of the rainfall, besides periodic inspections of the territory, could be parts of an early debris warning system, to be calibrated according to the specificity of the basin under analysis and validated through more historical cases as well.

**Author Contributions:** Conceptualization, A.P.; data curation, L.T.; formal analysis, A.P., J.C., L.T. and N.S.; funding acquisition, A.P. and N.S.; investigation, A.P. and J.C.; methodology, A.P. and J.C.; project administration, N.S.; software, A.P. and J.C.; supervision, A.P.; validation, A.P.; visualization, L.T.; writing—review and editing, A.P. and N.S. Each author contributed equally to the first version of the manuscript, which was reviewed and approved by all the co-authors before submission. All authors have read and agreed to the published version of the manuscript.

**Funding:** This work was supported by the research' funds supplied by G. D'Annunzio University of Chieti-Pescara.

**Institutional Review Board Statement:** Not applicable.

**Informed Consent Statement:** Not applicable.

**Data Availability Statement:** Not applicable.

**Acknowledgments:** This work was supported by the research' funds supplied by G. D'Annunzio University of Chieti-Pescara. The authors would like to thank two anonymous referees for their effort and appreciations.

**Conflicts of Interest:** The authors declare no conflict of interest.

## Appendix A

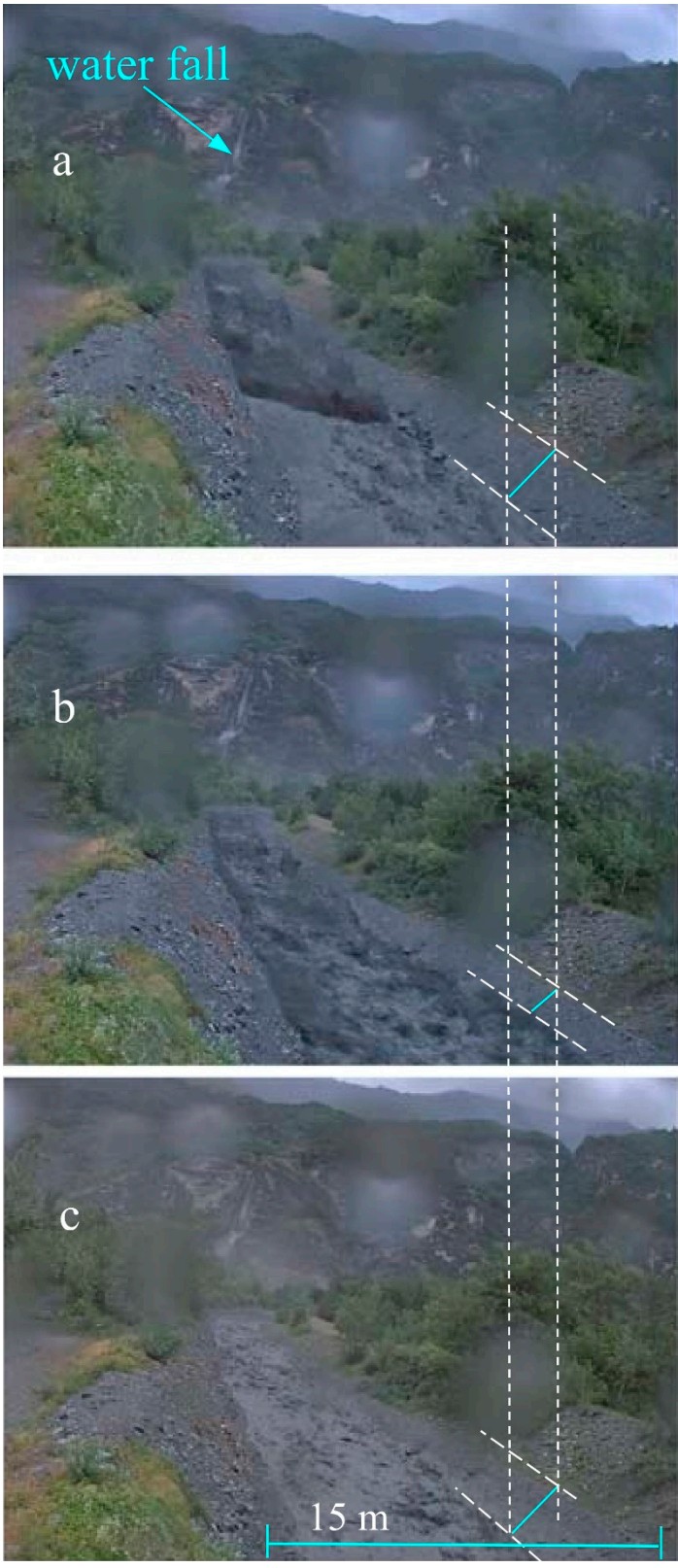

**Figure A1.** Frames from the movie of the studied event; (**a**) few seconds before the first spike arrival; (**b**) during spikes; (**c**) few seconds after the last spike.

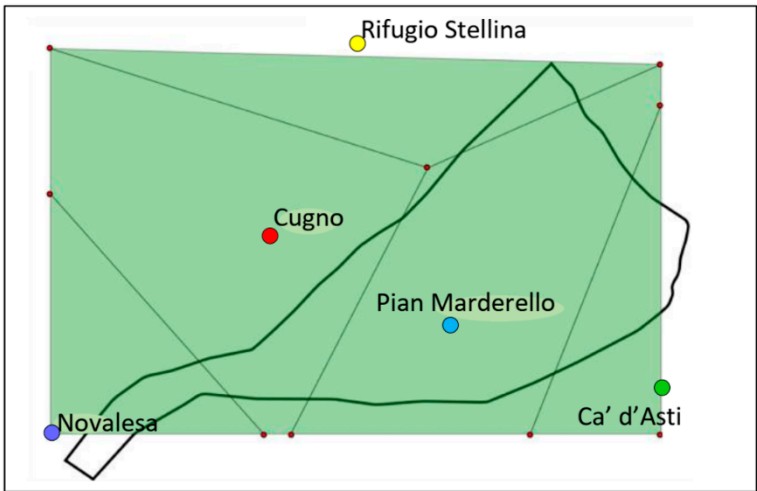

**Figure A2.** Thiessen's (Voronoi) polygonization of the Marderello's basin.

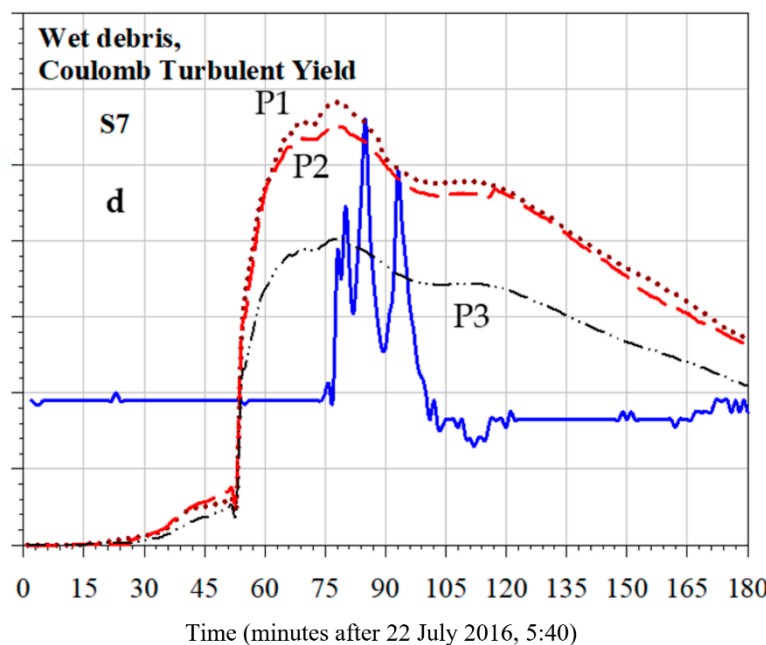

**Figure A3.** Simulation S7.

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
