# Peer review of "Learning Case Study of a Shallow-Water Model to Assess an Early-Warning System for Fast Alpine Muddy-Debris-Flow"

_water, doi:10.3390/w13060750_

Round 1

Reviewer 1 Report

In the current high-performance computing (HPC) context, where accelerated computing is revolutionizing not only the computation capacity but also the computation time required to perform the numerical simulation of large problems, there are interesting applications to real cases as the present paper. Authors present a work where the RiverFlow2D code is used. This is a 2D shallow-water model (based on Murillo & García-Navarro 2010) with several rheological options, as the mud and debris flow model or the Coulomb-Turbulent-Yield module between others. The numerical model solves the SW- equations using a Finite Volume numerical scheme applied over an unstructured mesh that results in a first order explicit Godonuv method that guaranties the conservation of the magnitudes that are simulated.  

The work is divided in 4 sections. The first section is an introduction where authors describe the problem and the way they are going to tackle it. The second section is devoted to show firstly, the geological and geographical settings, in situ used devices and the measured data. Secondly, they introduce the numerical approach (RiverFlow2D model) and the different rheological capabilities of the model that they are going to use to try to solve the problem. In the third section results are shown and discussed. The paper finishes with the conclusions section and a list of 45 references.

My recommendation is to accept after some revisions. 

My assessment is included in the attached file. 

Reviewer 2 Report

I think that this manuscript could constitute an important research contribution in the topic of early warning systems.

However, some crucial points could be mentioned into the introduction and in the conclusions parts, in order to provide a complete overview of this topic.

First of all, the title refers to early warning systems (EWS), but into the introduction no reference about classical schemes for EWSs is mentioned, schemes which are usually based on rainfall thresholds  that can be discriminated in deterministic and probabilistic (see De Luca and Versace, 2017).

Moreover, as authors proposed a physically based methodology for an EWS, they could discuss in the introduction the advantages of using physically based models with respect to an empirical model (i.e., as examples,  Intensity-Duration schemes for rainfall), in terms of understanding triggering mechanisms since they attempt to reproduce the physical behavior of the processes involved at hillslope scale, using detailed hydrological, hydraulic and geotechnical information (Baum et al., 2002; Rigon et al., 2006).

Finally, in their conclusions, authors could focus on the possibility of using their methodology also for calibration of a classical early warning system, based on rainfall thresholds, if a consistent number of historical events is available for a specific case study.

References to be cited:

  1. Baum, R. L., Savage, W. Z., and Godt, J. W: TRIGRS-A Fortran program for transient rainfall infiltration and grid-based regional slope-stability analysis, version 2.0, US Geological Survey Open-File Report 2008–1159, available at: http://pubs.usgs. gov/of/2008/1159/, 2008.
  2. De Luca, D. L. and Versace, P.: Diversity of Rainfall Thresholds for early warning of hydro-geological disasters, Adv. Geosci., 44, 53–60, https://doi.org/10.5194/adgeo-44-53-2017, 2017.
  3. Rigon, R., Bertoldi, G., and Over, T. M.: GEOtop: A Distributed Hydrological Model with Coupled Water and Energy Budgets, J. Hydrometeorol., 7, 371–388, 2006.

Round 2

Reviewer 1 Report

After the changes carried out by authors I think that the paper presents  a very good form and can be accepted in the present form.